# Weak-to-Strong Search:
# Align Large Language Models via
# Searching over Small Language Models

**Zhanhui Zhou**[*†]**, Zhixuan Liu**[*]**, Jie Liu, Zhichen Dong, Chao Yang**[†]**, Yu Qiao**
Shanghai Artificial Intelligence Laboratory
[*]Core Contribution, [†]Corresponding Author
asap.zzhou@gmail.com, yangchao@pjlab.org.cn
Code: https://github.com/ZHZisZZ/weak-to-strong-search

## Abstract

Large language models are usually fine-tuned to align with human preferences. However, fine-tuning a large language model can be challenging. In this work, we introduce *weak-to-strong search*, framing the alignment of a large language model as a test-time greedy search to maximize the log-probability difference between small tuned and untuned models while sampling from the frozen large model. This method serves both as (1) a compute-efficient model up-scaling strategy that avoids directly tuning the large model and as (2) an instance of weak-to-strong generalization that enhances a strong model with weak test-time guidance. Empirically, we demonstrate the flexibility of weak-to-strong search across different tasks. In controlled-sentiment generation and summarization, we use tuned and untuned gpt2s to improve the alignment of large models without additional training. Crucially, in a more difficult instruction-following benchmark, AlpacaEval 2.0, we show that reusing off-the-shelf small models (e.g., zephyr-7b-beta and its untuned version) can improve the length-controlled win rates of both white-box and black-box large models against gpt-4-turbo (e.g., $34.4\% \rightarrow 37.9\%$ for Llama-3-70B-Instruct and $16.0\% \rightarrow 20.1\%$ for gpt-3.5-turbo-instruct), despite the small models' low win rates $\approx 10.0\%$.

## 1 Introduction

Learning-based algorithms [1, 2, 3, 4, 5] have become the standard approach for aligning large language models (LLMs) with human preferences [3, 6, 7, 8, 9, 10]. However, fine-tuning large language models is resource-intensive and difficult to implement [4]. These challenges have motivated recent studies on search-based algorithms that keep the large language models frozen and steer their decoding with test-time guidance [11, 12, 13, 14, 15, 16, 17]. Typical examples of search-based algorithms include rejection sampling [16, 17] and Monte Carlo Tree Search [18, 19]. These search-based algorithms are promising as they can reuse the same guiding signal to steer the decoding of any large language model without additional training. However, existing search-based methods either simplify the search over tokens as a bandit problem [16, 17], which limits their steerability, or require a value function learned from scratch to address preference reward sparsity and prune search space [13, 18, 14], which can be as difficult as fine-tuning a large language model.

To make search-based algorithms better suited for aligning large language models, we introduce *weak-to-strong search*, a simple algorithm that frames the alignment of a large model as a test-time search over the log-probabilities of small language models. This algorithm makes two contributions: **(1)** First, it builds on the theoretical foundation of the token-level MDP for alignment [20], using the

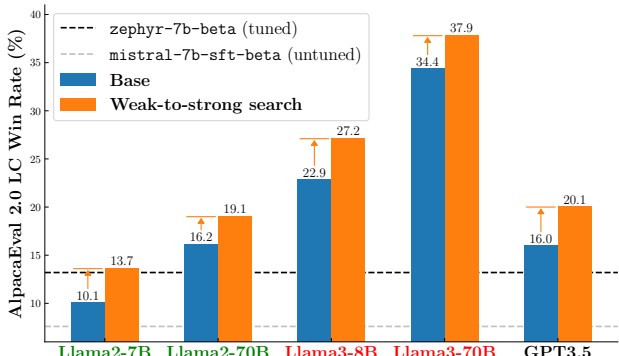

Figure 1: Weak-to-strong search enhances the alignment of large models through test-time guidance from small models (dashed lines). This method is applicable to white-box models that use the **same** or **different** vocabularies as the small models, as well as to **black-box** models. We present the results for the instruction-tuned models from each family (e.g., **Llama2-7B** denotes `Llama-2-7b-chat`).

log-probability difference between small tuned and untuned language models as both reward and value [4, 20] to guide the decoding of a large model (Section 4.1). Theoretically, this formulation is suitable for search as it converts the otherwise sequence-level sparse preference reward function to a per-token dense reward function, which can be summed up as a value function [20]. Practically, this formulation allows the reuse of off-the-shelf small tuned and untuned language model pairs as steering forces, avoiding the need to train a reward or value model from scratch. **(2)** Second, it introduces a beam search variant, Chunk-level Beam Search (CBS), tailored for optimizing the proposed search objective. CBS guides the large language model towards high-reward regions by alternating between sampling from the frozen large model and expanding promising states as evaluated by the small tuned and untuned models (Section 4.2). Especially, when the small models are weaker than the large model, our method can be viewed as an instance of weak-to-strong generalization [21] that makes the strong model stronger with weak test-time guidance (Figure 1).

Empirically, we verify weak-to-strong search's flexibility in various tasks (Section 5). First, in controlled-sentiment generation [22] and summarization [2], our method uses small language models of 124M parameters (i.e., `gpt2`) to effectively steer much larger language models from the GPT-2 (e.g., `gpt2-xl`) [23], Llama-2 [7] and Llama-3 [24] families, at least as effective as existing methods. Then, in a more difficult instruction-following benchmark, AlpacaEval 2.0 [25], we show reusing off-the-shelf small models (e.g., `zephyr-7b-beta` and its untuned version) as test-time guidance can significantly improve the length-controlled win rates of both white-box and black-box large models against `gpt-4-turbo` (e.g, $34.4\% \rightarrow 37.9\%$ for `Llama-3-70B-Instruct`, and $16.0\% \rightarrow 20.1\%$ for `gpt-3.5-turbo-instruct`), despite the small models' low win rates $\approx 10.0\%$ (Figure 1).

## 2 Related Work

Large unsupervised language models trained on internet-scale corpus acquire broad knowledge and abilities [26, 27, 28]. However, these large pre-trained language models may not always align with human values. To instill the desired behaviors into language models, most existing methods fine-tune these pre-trained language models on human comparisons of model-generated responses [1, 2, 3, 4, 7, 24, 6, 29]. Despite these successes, fine-tuning a large language model requires substantial computational resources and engineering effort. These problems are compounded by the reality that different humans have different values [3, 30, 13, 31, 32, 33], as it is nearly impossible to train a new large language model from scratch for individual preference. In light of these issues, our work takes a search-based approach, folding as much of the complexity of alignment as possible into the decoding phase. This allows us to keep the large pre-trained language models frozen, steering their outputs at test time with only small models that are easier to obtain.

Framing alignment as a test-time search to maximize a reward function is not a novel formulation. However, most existing works either simplify autoregressive decoding as a bandit problem [16, 17], which limits their steerability, or require a value function learned from scratch to handle sparse

preference rewards and prune search space [13, 18], which can be as difficult as training a large language model from scratch. Our work avoids these issues by parametrizing the sparse preference reward function with the log-probability difference between small tuned and untuned language models [4]. This parametrization not only simplifies the search objective, allowing a simple greedy search algorithm to generate good results, but also reuses off-the-shelf models as steering forces, eliminating the need to train a reward or critic model from scratch.

Concurrently with our work, Rafailov et al. [20] proposes a token-level MDP interpretation for language model alignment, demonstrating that a greedy probability search over a trained language model can achieve improvements over regular decoding. Our work builds on their theoretical foundations and proposes a practical greedy search algorithm designed for weak-to-strong guidance.

The idea of using small language models to align large language models has arisen in many recent works. The most related is proxy or emulated fine-tuning [34, 12, 11, 35], which uses the distributional difference of a small tuned and untuned model pair to modify the output distribution of a large model, approximating the output of the directly tuned large model. However, these methods require that both small and large models share the same vocabulary, limiting their practical applications. In contrast, our approach does not modify the sampling distribution of the large model at the token level. Instead, we perform a tree search that periodically prioritizes the most promising states for further expansion (as evaluated by the small models) while sampling from the frozen large model's distribution. Thus our approach does not require shared vocabulary and is applicable to black-box language models.

## 3 Preliminaries

In this section, we introduce the mathematical formulation of alignment (Section 3.1) and describe the duality between language models and reward functions (Section 3.2).

### 3.1 Aligning Language Models with Human Preferences

The alignment of language models is typically cast as a KL-constrained optimization problem [1]:

$$\arg\max_{\pi} \quad \mathbb{E}_{\mathbf{x}\sim p(\mathbf{x}),\mathbf{y}\sim\pi(\mathbf{y}|\mathbf{x})}\left[r(\mathbf{x},\mathbf{y})\right] \tag{1a}$$

$$\text{s.t.} \quad \mathbb{E}_{\mathbf{x}\sim p(\mathbf{x})}\left[\mathbb{D}_{\mathrm{KL}}\left(\pi(\mathbf{y}\mid\mathbf{x})\,\|\,\pi_{\mathrm{ref}}(\mathbf{y}\mid\mathbf{x})\right)\right] \leq \epsilon, \tag{1b}$$

where $p(\mathbf{x})$ is a distribution of prompts, $\mathbf{y}$ is the complete language model response, $r$ is a preference reward function that encourages human-preferred responses, and $\mathbb{D}_{\mathrm{KL}}$ limits how far the optimized language model $\pi$ can deviate from the reference (untuned) model $\pi_{\mathrm{ref}}$. There are two main categories of alignment algorithms: (1) *search-based* algorithms that optimize Eq. 1 with graph-based search during inference [16, 13, 18, 19, 11, 12], and (2) *learning-based* algorithms that optimize Eq. 1 through gradient descent, aiming for a parametrized optimal language model [1, 36, 4, 37]. Our work falls in the first category, proposing a search-based algorithm capable of using small language models to guide the decoding of a large language model to align with human preferences.

### 3.2 Duality between Language Models and Reward Functions

The analytical solution to Eq. 1 can be obtained through the following Lagrangian [38, 39]:

$$\mathcal{L}(\pi,\beta) = \mathbb{E}_{\mathbf{x}\sim p(\mathbf{x}),\mathbf{y}\sim\pi(\mathbf{y}|\mathbf{x})}\left[r(\mathbf{x},\mathbf{y}) + \beta\left(\epsilon - \mathbb{D}_{\mathrm{KL}}\left(\pi(\mathbf{y}\mid\mathbf{x})\,\|\,\pi_{\mathrm{ref}}(\mathbf{y}\mid\mathbf{x})\right)\right)\right], \tag{2}$$

which has a well-known closed-form solution that expresses a duality between the reward function $r(\mathbf{x},\mathbf{y})$ and the optimal language model $\pi^*(\mathbf{y}\mid\mathbf{x})$ [40, 41]:

$$r(\mathbf{x},\mathbf{y}) = \beta\log\frac{\pi^*(\mathbf{y}\mid\mathbf{x})}{\pi_{\mathrm{ref}}(\mathbf{y}\mid\mathbf{x})} + \beta\log Z(\mathbf{x}), \tag{3}$$

where $Z(\mathbf{x}) = \sum_{\mathbf{y}}\pi_{\mathrm{ref}}(\mathbf{y}\mid\mathbf{x})\exp\left(\frac{1}{\beta}r(\mathbf{x},\mathbf{y})\right)$ denotes the partition function. One takeaway from this duality is that we can always express a reward function using tuned and untuned language models: (1) If a reward function is given [1, 2, 3], we can first obtain the optimally tuned language model under this reward function with any learning-based algorithms, and then use the tuned and untuned models $(\pi^*, \pi_{\mathrm{ref}})$ to reparametrize the reward function [42]; (2) If a dataset is given from which the reward function can be derived, we can then directly parametrize the reward function with the tuned and untuned language models $(\pi^*, \pi_{\mathrm{ref}})$ during reward modeling [4].

# 4 Weak-to-Strong Search

In this section, we introduce weak-to-strong search, a search-based algorithm that aligns a large language model by searching over the log-probability difference between small tuned and untuned language models. First, we discuss how using language models to parametrize the preference reward function (Eq. 1) makes the reward-maximization problem solvable by a simple greedy search algorithm (e.g., beam search) (Section 4.1). Then, we introduce a practical beam search method, Chunk-level Beam Search (CBS) (Section 4.2), that balances reward maximization and KL minimization, which is applicable to steering both white-box and black-box large language models.

## 4.1 Language Models as Both Reward and Value Functions

One practical challenge for search-based alignment algorithms is the sparsity of the preference reward signal. The preference reward function $r(\mathbf{x}, \mathbf{y})$, based on the Bradley-Terry model [43], only emits a terminal reward when the model response is complete. Search-based algorithms often struggle without any intermediate rewards or a value function providing intermediate guidance [44, 45]. However, if we parameterize this sparse reward function with language models (Section 3.2), we can obtain both a dense reward function and a value function simultaneously.

**Language models as a dense reward function.** To obtain a dense reward function, we leverage the duality between the sparse preference reward function and the dense language model probability (Eq. 3). By explicitly factorizing the log-probability of a complete response $\mathbf{y}$ under the language models, we obtain a sum-of-rewards style formulation for Eq. 3:

$$r(\mathbf{x}, \mathbf{y}) = \beta \left( \sum_{t=1}^{|\mathbf{y}|} \log \frac{\pi^*(\mathbf{y}_t \mid \mathbf{x}, \mathbf{y}_{<t})}{\pi_{\text{ref}}(\mathbf{y}_t \mid \mathbf{x}, \mathbf{y}_{<t})} \right) + \beta \log Z(\mathbf{x}), \tag{4}$$

where $\mathbf{y}_{<t}$ denotes the response tokens from $1$ to $t-1$, and the last response token $\mathbf{y}_{|\mathbf{y}|}$ is the EOS token. Combining Eq. 1 and 4, we rewrite the original objective with a per-token reward function:

$$\underset{\pi}{\arg\max} \quad \mathbb{E}_{\mathbf{x} \sim p(\mathbf{x}), \mathbf{y} \sim \pi(\mathbf{y}|\mathbf{x})} \left[ \sum_{t=1}^{|\mathbf{y}|} \log \frac{\pi^*(\mathbf{y}_t \mid \mathbf{x}, \mathbf{y}_{<t})}{\pi_{\text{ref}}(\mathbf{y}_t \mid \mathbf{x}, \mathbf{y}_{<t})} \right] \tag{5a}$$

$$\text{s.t.} \quad \mathbb{E}_{\mathbf{x} \sim p(\mathbf{x})} \left[ \mathbb{D}_{\text{KL}} \left( \pi(\mathbf{y} \mid \mathbf{x}) \,\|\, \pi_{\text{base}}(\mathbf{y} \mid \mathbf{x}) \right) \right] \le \epsilon, \tag{5b}$$

where $\beta$ and $Z(\mathbf{x})$ are omitted as they do not influence the optimal solution. It is important to note that the reference model that parametrizes the reward function ($\pi_{\text{ref}}$) (Eq. 5a) and the reference model that constrains the test-time search space ($\pi_{\text{base}}$) (Eq. 5b) can be different. **Practically, decoupling the reference models is useful as it allows using a tuned and untuned language model pair - namely $(\pi^*, \pi_{\text{ref}})$ - to steer the decoding of any base language model $\pi_{\text{base}}$ without retraining**.

Setting aside the KL constraint (Eq. 5b) for now, we can apply existing search algorithms like beam search [45, 20] to optimize Eq. 5a. Beam search is often criticized for leading to myopic solutions [45], as it tends to greedily prioritize states $(\mathbf{x}, \mathbf{y}')$ ($\mathbf{y}'$ is incomplete)[1] with high cumulative reward $\log \pi^*(\mathbf{y}' \mid \mathbf{x}) - \log \pi_{\text{ref}}(\mathbf{y}' \mid \mathbf{x})$ midway through generation, which is generally viewed as poorly correlated with the overall return we care about. While this criticism is valid for most MDPs, we argue that in the token-level MDP [20] of our case, the cumulative reward mid-generation is actually a reliable indicator of the long-term value, making beam search less myopic.

**Cumulative reward under language models as a value function [20].** Appendix A shows that:

$$\log \frac{\pi^*(\mathbf{y}' \mid \mathbf{x})}{\pi_{\text{ref}}(\mathbf{y}' \mid \mathbf{x})} \propto \begin{cases} -V^*(\mathbf{x}) + V^*(\mathbf{x}, \mathbf{y}') & \text{if } \mathbf{y}' \text{ is incomplete} \\ -V^*(\mathbf{x}) + r(\mathbf{x}, \mathbf{y}') & \text{if } \mathbf{y}' \text{ is complete}, \end{cases} \tag{6}$$

where $V^*(\mathbf{x}, \mathbf{y}')$ denotes the value function, predicting the expected terminal reward under the optimal $\pi^*$ in the original KL-constrained sparse reward problem. Although $V^*(\mathbf{x}, \mathbf{y}')$ is not necessarily achievable by the searched policy, it approximates how good the state $(\mathbf{x}, \mathbf{y}')$ is in the long run. In other words, continuing from the state $(\mathbf{x}, \mathbf{y}')$ of high cumulative reward $\log \pi^*(\mathbf{y}' \mid \mathbf{x}) - \log \pi_{\text{ref}}(\mathbf{y}' \mid \mathbf{x})$ is likely to generate a complete response $\mathbf{y}$ with high overall return $\log \pi^*(\mathbf{y} \mid \mathbf{x}) - \log \pi_{\text{ref}}(\mathbf{y} \mid \mathbf{x})$.

---

[1] $\mathbf{y}$ denotes a complete response, while $\mathbf{y}'$ denotes a response that can be either incomplete or complete.

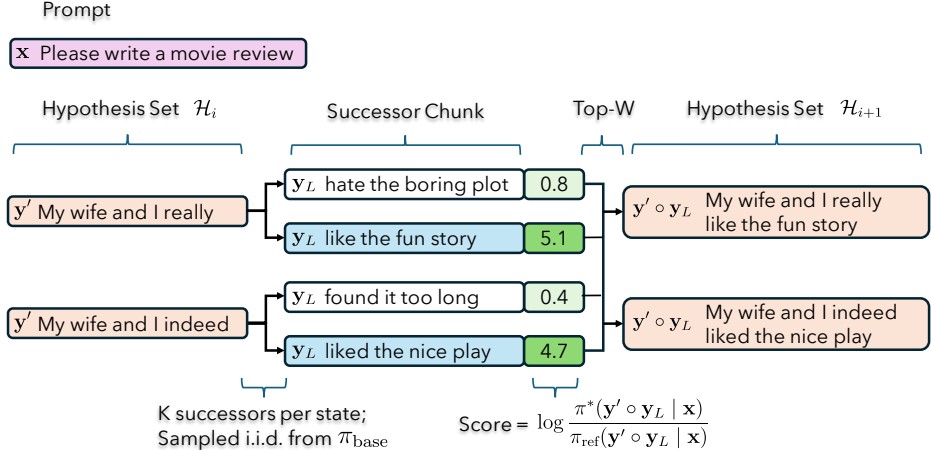

Figure 2: Illustration of Chunk-level Beam Search with $W, K = 2, 2$.

## 4.2 Chunk-level Beam Search (CBS)

After analyzing the feasibility of optimizing Eq. 5a with greedy search algorithms (e.g., beam search), we introduce a practical beam search variant that optimizes the dense reward objective (Eq. 5a) while ensuring the KL-constraint from $\pi_{\text{base}}$ (Eq. 5b).

The core algorithm providing the foundation of our method, Chunk-level Beam Search (CBS), is detailed in Algorithm 1 and illustrated in Figure 2. The key insight is that our beam search operates at the level of chunk. The search starts at the prompt and always maintains a hypothesis set $\mathcal{H} = \{(\mathbf{x}, \mathbf{y}')_i\}_{i=1}^{W}$ of $W$ states. For each state $(\mathbf{x}, \mathbf{y}')$ in $\mathcal{H}$, CBS samples $K$ continuation chunks $\mathbf{y}_L$ of length $L$ from $\pi_{\text{base}}$. This results in $WK$ successor states. Among these successors, only the top-$W$ successors with the highest partial return $\log \pi^*(\mathbf{y}' \circ \mathbf{y}_L \mid \mathbf{x}) - \log \pi_{\text{ref}}(\mathbf{y}' \circ \mathbf{y}_L \mid \mathbf{x})$ are stored in $\mathcal{H}$ and expanded further. Finally, the terminal state $(\mathbf{x}, \mathbf{y})$ with the highest intermediate return $\log \pi^*(\mathbf{y} \mid \mathbf{x}) - \log \pi_{\text{ref}}(\mathbf{y} \mid \mathbf{x})$ is selected, from which the complete response $\mathbf{y}$ is extracted.

---

**Algorithm 1** Chunk-level Beam Search (CBS)

---

1: **Input:** prompt $\mathbf{x}$, beam width $W$, successors per state $K$, chunk length $L$,
2:          model to steer $\pi_{\text{base}}$, tuned model $\pi^*$, and untuned model $\pi_{\text{ref}}$.
3: **Output:** optimal terminal state $(\mathbf{x}, \mathbf{y})$
4: Initialize $\mathcal{H} = \{(\mathbf{x}, \mathbf{y}' = \varnothing)_i\}_{i=1}^{W}$
5: **while** $\exists (\mathbf{x}, \mathbf{y}') \in \mathcal{H}$ such that $\mathbf{y}'$ is incomplete **do**
6:      Initialize $\mathcal{C} = \{\}$
7:      **for** each $(\mathbf{x}, \mathbf{y}') \in \mathcal{H}$ **do**
8:          $\mathcal{Y} \leftarrow \{(\mathbf{y}_L)_i\}_{i=1}^{K} \overset{\text{i.i.d.}}{\sim} \pi_{\text{base}}(\cdot \mid \mathbf{x}, \mathbf{y}')$                      // $\mathbf{y}_L = \varnothing$ if $\mathbf{y}'$ is complete
9:          $\mathcal{C} \leftarrow \mathcal{C} \cup \{(\mathbf{x}, \mathbf{y}' \circ \mathbf{y}_L) \mid \mathbf{y}_L \in \mathcal{Y}\}$
10:      **end for**
11:      $\mathcal{H} \leftarrow \text{Top-}W_{(\mathbf{x}, \mathbf{y}' \circ \mathbf{y}_L) \in \mathcal{C}} (\log \pi^*(\mathbf{y}' \circ \mathbf{y}_L \mid \mathbf{x}) - \log \pi_{\text{ref}}(\mathbf{y}' \circ \mathbf{y}_L \mid \mathbf{x}))$
12: **end while**
13: **return** $\arg\max_{(\mathbf{x}, \mathbf{y}) \in \mathcal{H}} (\log \pi^*(\mathbf{y} \mid \mathbf{x}) - \log \pi_{\text{ref}}(\mathbf{y} \mid \mathbf{x}))$

---

**CBS is a unified framework that encompasses several search-based algorithms**: (1) CBS with $W = 1$, $K = N$, $L = \infty$ (i.e., infinite chunk length) is equivalent to BoN sampling with $\log \pi^*(\mathbf{y} \mid \mathbf{x}) - \log \pi_{\text{ref}}(\mathbf{y} \mid \mathbf{x})$ as the scoring function, and (2) CBS with $K = \infty$, $L = 1$ (i.e., exploring all possible next tokens from the vocabulary) is equivalent to vanilla token-level beam search. *However, we always ensure finite chunk length and limited successor exploration via sampling to achieve the best of both worlds*: (1) Using a finite chunk length allows CBS to prune bad states during generation, enhancing steerability more efficiently compared to BoN. (2) Sampling from

$\pi_{\text{base}}$ with limited successor exploration implicitly enforces the KL-constraint from $\pi_{\text{base}}$ (Eq. 5b); otherwise, integrating the KL-constraint into the objective (Eq. 5a) would be necessary for token-level search, but this can be challenging, especially when vocabularies of models differ or with black-box language base models $\pi_{\text{base}}$ whose log-probabilities are inaccessible.

**Computation costs.** In practice, CBS samples $WK$ continuation chunks in parallel from the frozen base model $\pi_{\text{base}}$ and prune states by calling tuned and untuned model pair $(\pi^*, \pi_{\text{ref}})$ every $L$ tokens. Larger $WK$ and smaller $L$ enhance steerability at the cost of increased computations. Note that high steerability, while beneficial, is not always ideal as it may lead to large KL deviation and over-optimization [16].

### 4.3 Application: Model Up-Scaling and Weak-to-Strong Generalization

The most practical use of CBS occurs when the tuned and untuned models, $(\pi^*, \pi_{\text{ref}})$, are smaller than the model to steer, $\pi_{\text{base}}$. (1) First, this instance serves as a model up-scaling strategy, directly tuning a small model $\pi_{\text{ref}} \to \pi^*$, by which the large model decoding can then be guided, to achieve similar outcomes as directly tuning the large model. (2) Second, since the small models $(\pi^*, \pi_{\text{ref}})$ are usually weaker than the large model to steer $\pi_{\text{base}}$, this instance also exemplifies weak-to-strong generalization [21], enhancing the strong model with only weak test-time guidance. We refer to this instance of CBS as weak-to-strong search, which is the main focus of our study.

## 5 Experiments

In this section, we empirically evaluate weak-to-strong search's ability to align large language models using only test-time guidance from small language models. First, in **controlled-sentiment generation** [22] and **summarization** [2], we tune gpt2 to model the desired behaviors in each task and then use tuned and untuned gpt2 to steer larger models of various scales (Section 5.1). Next, in a more difficult **instruction-following** benchmark, AlpacaEval 2.0 [25], instead of tunning small models, we reuse off-the-shelf open-source 7B models and their untuned versions to steer a series of large models, including open-source 70B models and a black-box model (Section 5.2).

**Baselines.** In addition to weak-to-strong search, we evaluate several existing test-time approaches that steer a large language model $\pi_{\text{base}}$ using small tuned and untuned language models $(\pi^*, \pi_{\text{ref}})$: (1) **Base**: we explore regular decoding from the frozen large language model with n-shot prompting (see Appendix B.1.6 for prompt details). (2) **Best-of-N Sampling (BoN)** [16, 17]: BoN uses $r = \log \pi^*(\mathbf{y} \mid \mathbf{x}) - \log \pi_{\text{ref}}(\mathbf{y} \mid \mathbf{x})$ to select the highest-scoring responses among the $N$ independent responses from the frozen large language model. Since weak-to-strong search (CBS) samples $WK$ response chunks in parallel, for fair computational comparisons, we always ensure $N = WK$. (3) **Emulated Fine-Tuning (EFT)** [34, 12, 11, 35]: EFT approximates the results of directly fine-tuning the large language model by sampling from $\log \pi_{\text{EFT}}(\mathbf{y}_t \mid \mathbf{x}, \mathbf{y}_{<t}) \propto \log \pi_{\text{base}}(\mathbf{y}_t \mid \mathbf{x}, \mathbf{y}_{<t}) + \beta^{-1}(\log \pi^*(\mathbf{y}_t \mid \mathbf{x}, \mathbf{y}_{<t}) - \log \pi_{\text{ref}}(\mathbf{y}_t \mid \mathbf{x}, \mathbf{y}_{<t}))$, where $\beta$ is the hyperparameter from Eq. 2. Note that EFT is only applicable when all models share the same vocabulary (which is necessary for composing output distributions from different models). Whenever possible, we also compare test-time methods against **directly fine-tuning** the large models in the same way small models are tuned.

### 5.1 Controlled-Sentiment Generation & Summarization

**Setup.** For these two tasks, we follow the synthetic setups from [16, 46, 4], assuming access to a gold reward model $r_{\text{gold}}$. For controlled-sentiment generation, $r_{\text{gold}}$ encourages positive continuations of movie reviews, while for summarization, it encourages high-quality summaries of Reddit posts (details in Appendix B.1.4). We generate synthetic preference datasets $\mathcal{D} = \{(\mathbf{x}, \mathbf{y}_w, \mathbf{y}_l)_i\}_{i=1}^N$ from $r_{\text{gold}}$ with $p(\mathbf{y}_1 \succ \mathbf{y}_2 \mid \mathbf{x}) = \sigma(r_{\text{gold}}(\mathbf{x}, \mathbf{y}_1) - r_{\text{gold}}(\mathbf{x}, \mathbf{y}_2))$ to mimic human feedback [47].

To obtain the small language models, we optimize gpt2 (124M parameters) using the standard DPO pipeline [4]: (1) we first obtain the reference model $\pi_{\text{ref}}$ through supervised fine-tuning on both chosen and rejected responses from the synthetic preference dataset, then (2) we apply DPO on the synthetic preference dataset with $\pi_{\text{ref}}$ as the reference policy to obtain the optimal language model $\pi^*$. Note that the first stage primarily informs the language model of the desired response format, with most of the tuning occurring in the second DPO stage.

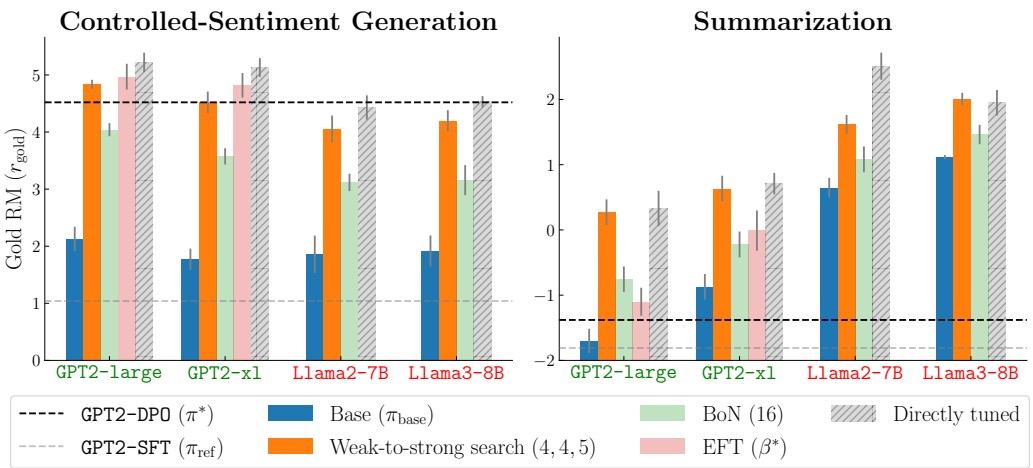

Figure 3: **The gold reward achieved for different large pre-trained models under the gpt2 guidance.** We show the mean reward ($\pm$ standard deviations) across three random seeds. EFT ($\beta^*$) denotes the best EFT results among $\beta \in \{1/4, 1/2, 1, 2, 4\}$; Weak-to-strong search $(4, 4, 5)$ denotes CBS with $W, K, L = 4, 4, 5$; BoN (16) denotes BoN with $N = 16$.

Given the tuned and untuned (un-DPO-tuned) gpt2 pair $(\pi^*, \pi_{\text{ref}})$, we use them to steer the large pre-trained language models without additional training. The large pre-trained language models we study fall into two categories based on whether they share the same vocabulary as the small models: (1) **same vocabulary**: gpt2-large (774M), gpt2-xl (1.5B) and (2) **cross vocabulary**: Llama-2-7b, Llama-3-8B. Eventually, since we have access to the gold reward model, language model responses can be fairly evaluated on the test split of prompts using this gold reward model.

**Results.** Figure 3 demonstrates weak-to-strong search's great flexibility and steerability in both tasks. For summarization, weak-to-strong search consistently outperforms other test-time methods by large margins. For controlled-sentiment generation, weak-to-strong search is second only to EFT with a carefully selected hyperparameter ($\beta^* = 1/4$) when EFT is applicable. We hypothesize that token-level adjustments from EFT are sufficient for controlled-sentiment generation, which primarily requires minor stylistic changes at the token level (e.g., "hate" $\rightarrow$ "love"). However, in the more complex task of summarization, where broader subsequence-level manipulations are essential, weak-to-strong search excels. Please refer to Appendix D for quantitative comparisons of samples from different methods. We need to mention that we do not meaningfully tune weak-to-strong search (CBS)'s hyperparameters to obtain the results in Figure 3 (we use a fixed set of hyperparameters of $(4, 4, 5)$ for $W, K, L$ across all models), which may underestimate the performance of our method. In addition, our method enables **consistent weak-to-strong generalization in the harder task of summarization**: most large pre-trained models (except for gpt2-large) are stronger than the tuned gpt2 in summarizing long text, but the weak models are still able to improve the strong models through test-time guidance, **nearly matching the results of direct fine-tuning**. The phenomenon of weak-to-strong generalization will be further studied in Section 5.2.

**Chunk-level Beam Search ablations.** We perform additional ablations to understand how CBS hyperparameters (beam width $W$, successors per state $K$, and chunk length $L$) influence performance. Figure 4 displays the ablation results for $W$ and $K$. With the same computation budget (i.e., $WK$), the optimal trade-off between $W$ and $K$ varies by tasks: for controlled-sentiment generation, the best results come from retaining the most promising state and concentrating computational efforts on expanding from it $(W, K = 1, 16)$; in contrast, for summarization, maintaining multiple hypotheses $(W, K = 8, 2)$ yields the best results probably because it helps avoid local optima. Figure 5 displays the ablation results for $L$ where smaller $L$ benefits controlled-sentiment generation, while an intermediate $L$ is optimal for summarization. These results are consistent with our findings from Figure 3, suggesting that the simple nature of controlled-sentiment generation makes token-level manipulation sufficient and cumulative reward mid-generation a more reliable indicator of overall return. See Appendix C.1 for extended ablations on more models.

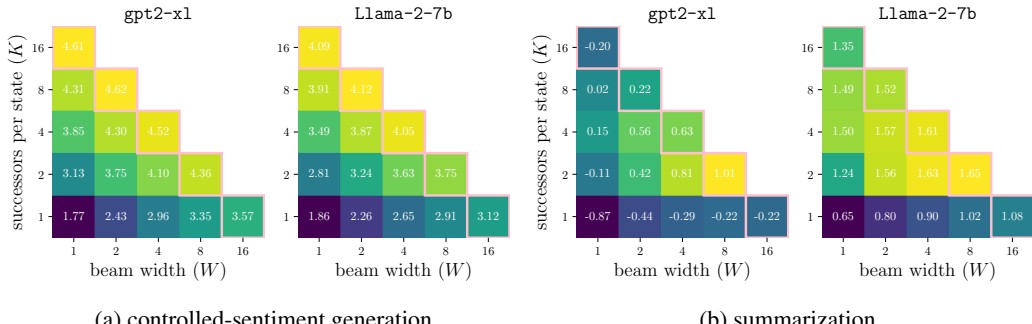

(a) controlled-sentiment generation

(b) summarization

Figure 4: **W, K ablations for CBS (L = 5).** We show the mean rewards across three random seeds. With the same computation budget (i.e., same $WK$), the optimal hyperparameters differ by tasks.

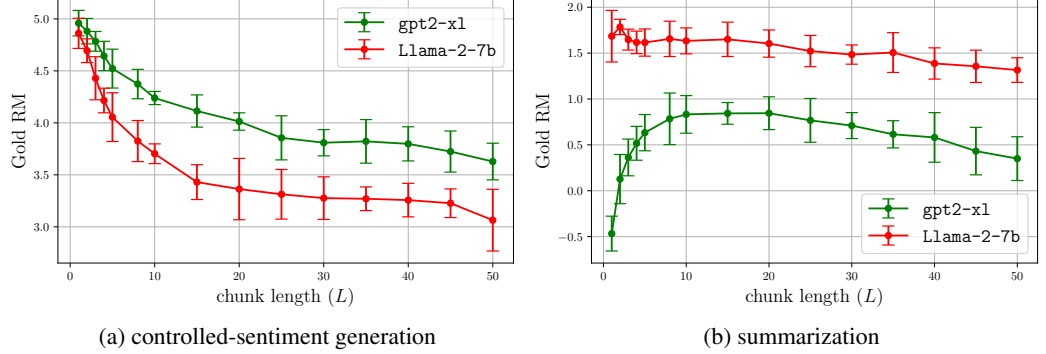

(a) controlled-sentiment generation

(b) summarization

Figure 5: **L ablations for CBS (W, K = 4, 4).** We show the mean rewards ($\pm$ standard deviations) across three random seeds.

## 5.2 Instruction Following

**Setup.** Next, we evaluate weak-to-strong search on a standard single-turn instruction-following benchmark, AlpacaEval 2.0 [25], which consists of 805 prompts from various open-source datasets. Unlike the previous section where we steer large *pre-trained* language models (e.g., `Llama-2-7b`), we now steer large *instruction-tuned* language models (e.g., `Llama-2-7b-chat`). This is because (1) instruction-tuned models often require further alignment to match human preferences [48], and (2) to study weak-to-strong generalization in instruction-following, the models must be proficient at following instructions before steering.

For small language models, we reuse two high-ranking 7B model pairs from the AlpacaEval 2.0 leaderboard as guidance: (1) **Zephyr guidance**: `zephyr-7b-beta` and its untuned version `mistral-7b-sft-beta`; (2) **Tulu guidance**: `tulu-2-dpo-7b` and its untuned version `tulu-2-7b`. All four models use the Llama-2 tokenizer. The large instruction-tuned language models we aim to further align fall into three categories: (1) **same vocabulary**: `Llama-2-7b-chat`, `Llama-2-70b-chat`; (2) **cross vocabulary**: `Llama-3-8B-Instruct`, `Llama-3-70B-Instruct`; and (3) **black box**: `gpt-3.5-turbo-instruct`. As it is nearly impossible to reproduce the exact training pipeline for these small models ($\pi_{\text{ref}} \to \pi^*$), we do not test the baseline results of directly fine-tuning the large models as in Figure 3. Language model responses are evaluated by their length-controlled win rates (LC WR) against `gpt-4-turbo`, with `gpt-4-turbo` serving as the judge.

**Results.** Experimental results with Zephyr and Tulu guidance are shown in Figure 6 (detailed hyperparameters in Appendix B.2.2). Weak-to-strong search consistently outperforms other test-time baselines with great margins. There are two crucial takeaways worth mentioning: **(1) Weak-to-strong search makes strong models stronger with only weak test-time guidance.** Take Zephyr guidance for an example (Figure 6, left), even if most large instruction-tuned models $\pi_{\text{base}}$ are stronger than `zephyr-7b-beta` before steering, weak-to-strong search is still able to enhance their performances

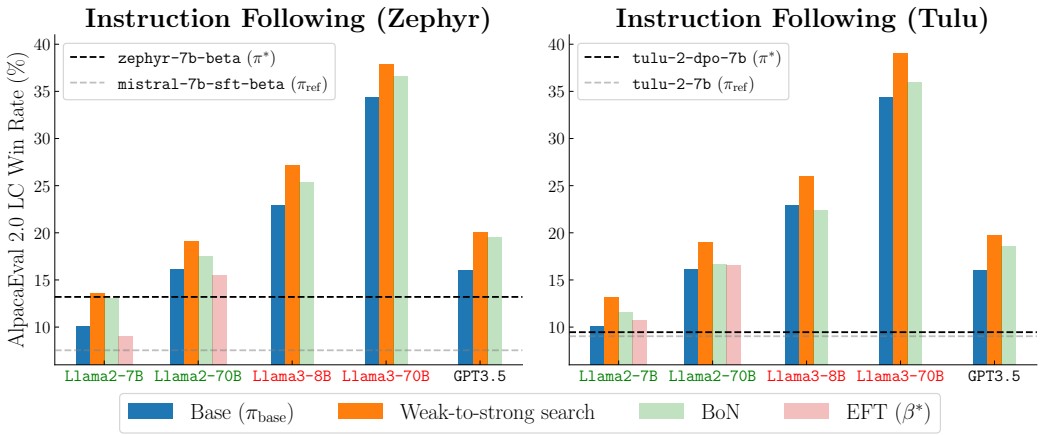

Figure 6: **The length-controlled win rates against gpt-4-turbo for various instruction-tuned models under Zephyr (left) or Tulu (right) guidance.** Hyperparameters are in Appendix B.2.2.

using weak models as guidance. Conversely, EFT and BoN mainly interpolate between weak and strong models, resulting in limited, if any, improvements over the strong models. We also tested beam search over the strong models without external guidance [20] but we found no obvious improvements compared with regular decoding (Table 2), probably because the latent reward functions behind these language models are not well aligned with the human preference that `gpt-4-turbo` approximates. The same observations apply to Tulu guidance, even though the tuned `tulu-2-dpo-7b` is weaker than all the large instruction-tuned language models by significant margins (Figure 6, right). **(2) Weak-to-strong search applies to black-box language models.** Our method, requiring only sampling from large language models, is also effective for black-box models like `gpt-3.5-turbo-instruct`. For weak-to-strong search with `gpt-3.5-turbo-instruct`, we use a relatively long chunk length of 100, as the black-box language model APIs are stateless and do not retain activation caches, making repeated context embedding costly. Despite the long chunk length, our method still effectively improves the alignment of black-box models, significantly outperforming BoN, a special case of weak-to-strong search (CBS) with infinite chunk length.

## 6 Discussion

We have presented weak-to-strong search, an alignment method that keeps the large language model frozen while steering its decoding through a test-time greedy search over small language models. This method builds on the insight that the log-probability difference between small tuned and untuned language models can serve both as a dense reward function and a value function, and then introduces a novel beam search algorithm designed for balancing reward maximization and KL minimization. This method offers a compute-efficient model up-scaling strategy that eliminates the complexity of directly fine-tuning the large models, and exemplifies weak-to-strong generalization [21] that makes strong models stronger with only weak test-time guidance. Empirically, this approach is effective in controlled-sentiment generation, summarization, and instruction following.

**Limitations & Future Work.** While our work focuses on aligning with human preferences, weak-to-strong search could also apply to tasks like reasoning [49, 50] and coding [51], where ground truth answers exist. This is because any pair of tuned and untuned language models can act as test-time steering forces, without necessarily being trained on preferences. This then raises several questions beyond the scope of our current study: (1) In our study, we consistently use SFTed policy as the untuned model $\pi_{\text{ref}}$ due to the two-stage nature of preference learning; however, in single-stage fine-tuning tasks, does weak-to-strong search still work with a pre-trained model serving as the untuned model $\pi_{\text{ref}}$? (2) Although our method shows consistent weak-to-strong generalization across diverse alignment tasks, it is also critical to probe its potential failure modes [52]. Can weak-to-strong search enhance language models in tasks where ground truth answers exist, beyond merely tailoring their knowledge and skills to human preferences? (3) Additionally, while our work mainly focuses on how dense language model reward function (Eq. 4) benefits language model decoding at test time, it's

also worth exploring the potential benefits of this reward parametrization for RL tuning. Although Appendix C.3 presents some promising preliminary results, we leave further analysis for future work.

## Acknowledgements

This work was supported in part by the National Key R&D Program of China (NO.2022ZD0160201). We would like to thank anonymous reviewers for their valuable feedback and helpful discussions.

## Author Contributions

**Zhanhui Zhou** led the project, proposed the research idea, wrote the codebase, designed the experiments, conducted most of the initial experiments, and wrote the paper. **Zhixuan Liu** assisted with running many of the ablation studies throughout the experiments presented in the paper. All other authors provided feedback throughout the project.

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

# A Mathematical Derivations for Eq. 6

As introduced in Section 3, the alignment objective under the traditional contextual bandit framing is given by (Eq. 2):

$$\arg\max_{\pi} \mathbb{E}_{\mathbf{x}\sim p(\mathbf{x}), \mathbf{y}\sim\pi(\mathbf{y}|\mathbf{x})} \left[ r(\mathbf{x}, \mathbf{y}) - \beta \mathbb{D}_{\mathrm{KL}}\left( \pi(\mathbf{y} \mid \mathbf{x}) \,\|\, \pi_{\mathrm{ref}}(\mathbf{y} \mid \mathbf{x}) \right) \right], \tag{7}$$

where $p(\mathbf{x})$ is a distribution of prompts, $\mathbf{y}$ is the complete language model response, $r$ is a reward function that encourages human-preferred responses, and $\mathbb{D}_{\mathrm{KL}}$ limits how far the optimized language model $\pi$ can deviate from the reference model $\pi_{\mathrm{ref}}$.

Given the autoregressive nature of language models, we can frame language model alignment as solving a token-level MDP [20]. This token-level MDP is define by the tuple $(\mathcal{S}, \mathcal{A}, f, r(\mathbf{s}_t, \mathbf{a}_t))$. Here, the state $\mathbf{s}_t := (\mathbf{x}, \mathbf{y}_{<t}) \in \mathcal{S}$ consists of the prompt and all response tokens generated so far; the action $\mathbf{a} := y_t$ determines the next token to generate from the vocabulary $\mathcal{A}$; the dynamics $f$ is a deterministic function that updates the state by concatenating the current state and action $\mathbf{s}_{t+1} := (\mathbf{s}_t, \mathbf{a}_t)$; $r(\mathbf{s}_t, \mathbf{a}_t)$ is a sparse reward that equals $r(\mathbf{x}, \mathbf{y})$ if $\mathbf{a}_t$ is EOS and 0 otherwise. We use $\rho_{\pi}(\mathbf{s}_t)$ to denote the state marginals of the trajectory distribution induced by the policy $\pi$. Under the token-level MDP, the objective from Eq. 7 can be written as

$$\max_{\pi} \sum_{t=1}^{T} \mathbb{E}_{\mathbf{s}_t\sim\rho_\pi(\mathbf{s}_t), \mathbf{a}_t\sim\pi(\mathbf{a}_t|\mathbf{s}_t)} \left[ r(\mathbf{s}_t, \mathbf{a}_t) + \underbrace{\beta \log \pi_{\mathrm{ref}}(\mathbf{a}_t \mid \mathbf{s}_t) + \beta\mathcal{H}(\pi(\mathbf{a}_t \mid \mathbf{s}_t))}_{-\beta\mathbb{D}_{\mathrm{KL}}[\pi(\mathbf{a}_t|\mathbf{s}_t)\,\|\,\pi_{\mathrm{ref}}(\mathbf{a}_t|\mathbf{s}_t)]} \right], \tag{8}$$

where $T$ specifies the response length. The solution of Eq. 8 is given by [40] as:

$$\pi^*(\mathbf{a}_t \mid \mathbf{s}_t) = \exp\left( \frac{1}{\beta} \left( Q^*(\mathbf{s}_t, \mathbf{a}_t) - V^*(\mathbf{s}_t) \right) \right), \tag{9}$$

where the optimal Q-function and V-function satisfies

$$Q^*(\mathbf{s}_t, \mathbf{a}_t) = r(\mathbf{s}_t, \mathbf{a}_t) + \beta \log \pi_{\mathrm{ref}}(\mathbf{a}_t \mid \mathbf{s}_t) + V^*(\mathbf{s}_{t+1}), \tag{10}$$

$$V^*(\mathbf{s}_t) = \sum_{i=t}^{T} \mathbb{E}_{\mathbf{s}_i\sim\rho_{\pi^*}(\mathbf{s}_i|\mathbf{s}_t), \mathbf{a}_i\sim\pi^*(\mathbf{a}_i|\mathbf{s}_i)} \left[ r(\mathbf{s}_i, \mathbf{a}_i) + \beta \log \pi_{\mathrm{ref}}(\mathbf{a}_i \mid \mathbf{s}_i) + \beta\mathcal{H}(\pi^*(\mathbf{a}_i \mid \mathbf{s}_i)) \right]. \tag{11}$$

Here, $V^*$ predicts the expected future return (the terminal reward in this sparse reward setting) penalized with the future KL constraint starting from the state $\mathbf{s}_t$, under the optimal language model $\pi^*$. Combining Eq. 9 and Eq. 10, we have

$$\beta \log \frac{\pi^*(\mathbf{a}_t \mid \mathbf{s}_t)}{\pi_{\mathrm{ref}}(\mathbf{a}_t \mid \mathbf{s}_t)} = r(\mathbf{s}_t, \mathbf{a}_t) + V^*(\mathbf{s}_{t+1}) - V^*(\mathbf{s}_t). \tag{12}$$

Note that (1) $r(\mathbf{s}_t, \mathbf{a}_t)$ is a sparse reward that is non-zero if $\mathbf{a}_t$ is EOS, and (2) $V^*(\mathbf{s}_{t+1}) = 0$ if $\mathbf{a}_t$ is EOS. Then, summing Eq. 12 from timestep 1 to $H$ yield

$$\sum_{t=1}^{H} \beta \log \frac{\pi^*(\mathbf{a}_t \mid \mathbf{s}_t)}{\pi_{\mathrm{ref}}(\mathbf{a}_t \mid \mathbf{s}_t)} = \begin{cases} -V^*(\mathbf{s}_1) + V^*((\mathbf{s}_H, \mathbf{a}_H)) & \text{if } \mathbf{a}_H \text{ is not EOS} \\ -V^*(\mathbf{s}_1) + r(\mathbf{s}_H, \mathbf{a}_H) & \text{if } \mathbf{a}_H \text{ is EOS,} \end{cases} \tag{13}$$

where $V^*(\mathbf{s}_{H+1}) = V^*((\mathbf{s}_H, \mathbf{a}_H))$ due to the deterministic transition. Now, returning back to the sequence-level MDP (Section 4), where we define $\mathbf{y}$ as a complete response, $\mathbf{y}'$ as a response that can be either complete or incomplete, we have that $\mathbf{s}_1 = (\mathbf{x}, \mathbf{y}_{<1}) = (\mathbf{x}, \varnothing) = \mathbf{x}$ and $(\mathbf{s}_H, \mathbf{a}_H) = ((\mathbf{x}, \mathbf{y}_{<H}), \mathbf{y}_H) = (\mathbf{x}, \mathbf{y}_{<H+1})$. Thus, we can rewrite Eq. 13, with a slight abuse of notations, as

$$\log \frac{\pi^*(\mathbf{y}' \mid \mathbf{x})}{\pi_{\mathrm{ref}}(\mathbf{y}' \mid \mathbf{x})} \propto \begin{cases} -V^*(\mathbf{x}) + V^*(\mathbf{x}, \mathbf{y}') & \text{if } y'_{|\mathbf{y}'|} \neq \text{EOS } (\mathbf{y}' \text{ is incomplete}) \\ -V^*(\mathbf{x}) + r(\mathbf{x}, \mathbf{y}') & \text{if } y'_{|\mathbf{y}'|} = \text{EOS } (\mathbf{y}' \text{ is complete}). \end{cases} \tag{14}$$

# B Further Details on the Experimental Setup

## B.1 Controlled-Sentiment Generation & Summarization

### B.1.1 Model Specification

The following table lists the models and their corresponding links.

| Models | Links |
|---|---|
| `gpt2` (124M) [23] | https://huggingface.co/openai-community/gpt2 |
| `gpt2-large` (774M) [23] | https://huggingface.co/openai-community/gpt2-large |
| `gpt2-xl` (1.5B) [23] | https://huggingface.co/openai-community/gpt2-xl |
| `Llama-2-7b` [7] | https://huggingface.co/meta-llama/Llama-2-7b-hf |
| `Llama-3-8B` [24] | https://huggingface.co/meta-llama/Meta-Llama-3-8B |

### B.1.2 Hyperparameters Specification

We use fixed hyperparameters across all tested models. We use temperature $T = 0.7$, top-k $= 50$ and top-p $= 1.0$ when sampling from the language models. For weak-to-strong search (CBS), we use $W, K, L = 4, 4, 5$ ($W$: beam width, $K$: successors per state, $L$: chunk length). For BoN, we use $N = 16$ for fair computational comparison with weak-to-strong search (i.e., $WK = N$). For EFT, we report the best results among $\beta \in \{1/4, 1/2, 1, 2, 4\}$.

### B.1.3 Compute Resources Specification

Models are evaluated over $1000$ test prompts, on one single NVIDIA A100 GPU.

### B.1.4 Gold Reward Model Details

We follow the synthetic setup in which we use the gold reward models to play the roles of humans and provide binary preference labels [16, 46, 4].

For controlled-sentiment generation, we reuse the publicly available `distilbert-imdb` to define the gold reward model $r_{\text{gold}}$. `Distilbert-imdb` is a fine-tuned classifier $p$ on the `imdb` dataset [53] to classify movie review sentiments. We define the gold reward $r_{\text{gold}}$ as $\log p(\text{positive} \,|\, x, y) - \log p(\text{negative} \,|\, x, y)$ to encourage positive review. Synthetic preferences are collected using the truncated movie reviews as prompts $\mathbf{x}$, and pairwise completions from `gpt2-imdb`, ranked with $p(\mathbf{y}_1 \succ \mathbf{y}_2 \,|\, \mathbf{x}) = \sigma(r_{\text{gold}}(\mathbf{x}, \mathbf{y}_1) - r_{\text{gold}}(\mathbf{x}, \mathbf{y}_2))$, as preferences.

For summarization, we fit a reward model on the `summarize_from_feedback` dataset [2] as the gold reward model $r_{\text{gold}}$. Specifically, this reward model is fine-tuned from `Llama-2-7b` with a linear projection head and binary cross entropy loss, using a batch size of $32$, a learning rate of `1e-5` for the projection head, and `5e-6` for other parameters, over one epoch with a cosine learning rate schedule. Synthetic preferences are generated by relabeling pairwise responses in the original dataset with $p(\mathbf{y}_1 \succ \mathbf{y}_2 \,|\, \mathbf{x}) = \sigma(r_{\text{gold}}(\mathbf{x}, \mathbf{y}_1) - r_{\text{gold}}(\mathbf{x}, \mathbf{y}_2))$.

Both gold reward models show high validation accuracies, $0.928$ and $0.736$, demonstrating strong correlation with human judgments.

### B.1.5 Direct Tuning Details

Direct tuning on the synthetic preferences $\mathcal{D} = \{(\mathbf{x}, \mathbf{y}_w, \mathbf{y}_l)_i\}_{i=1}^N$ involves two stages: Supervised Fine-Tuning (`SFT`) and Direct Preference Optimization (`DPO`) [4]. During `SFT`, models are trained on both selected and rejected responses using a batch size of $64$, a learning rate of `2e-5`, and a cosine learning rate schedule over one epoch. During `DPO`, we use a $\beta = 0.1$, batch size of $256$, a learning rate of `1e-6`, and a cosine learning rate schedule over one epoch.

### B.1.6 Prompt Template for Sampling from Base Models

When sampling from large pre-trained models, it is crucial to provide clear task-specific instructions. For sentiment-controlled generation, we use a zero-shot prompt:

```
Here is a movie review from imdb: {prompt}
```

For summarization, we use a two-shot prompt (the exemplars are selected arbitrarily):

```
{examplar[1].prompt}TL;DR: {examplar[1].response}
{examplar[2].prompt}TL;DR: {examplar[2].response}
{prompt}TL;DR:
```

### B.2 Instruction Following

#### B.2.1 Model Specification

The following table lists the models and their corresponding links.

| Models | Links |
|---|---|
| zephyr-7b-beta [9] | https://huggingface.co/HuggingFaceH4/zephyr-7b-beta |
| mistral-7b-sft-beta [9] | https://huggingface.co/HuggingFaceH4/mistral-7b-sft-beta |
| tulu-2-dpo-7b [54] | https://huggingface.co/allenai/tulu-2-dpo-7b |
| tulu-2-7b [54] | https://huggingface.co/allenai/tulu-2-7b |
| Llama-2-7b-chat [7] | https://huggingface.co/meta-llama/Llama-2-7b-chat-hf |
| Llama-2-70b-chat [7] | https://huggingface.co/meta-llama/Llama-2-70b-chat-hf |
| Llama-3-8B-Instruct [24] | https://huggingface.co/meta-llama/Meta-Llama-3-8B-Instruct |
| Llama-3-70B-Instruct [24] | https://huggingface.co/meta-llama/Meta-Llama-3-70B-Instruct |
| gpt-3.5-turbo-instruct [3] | https://platform.openai.com/docs/models/gpt-3-5-turbo |

#### B.2.2 Hyperparameters Specification

When sampling from Llama-3-8B-Instruct and Llama-3-70B-Instruct, we use the $T = 0.6$, top-k $= 1.0$ and top-p $= 0.9$ as per the official Llama-3 generation configuration. For other models, we default to temperature $T = 0.7$, top-k $= 50$ and top-p $= 1.0$. Specific hyperparameters for each method are detailed in Tables 3 and 4.

#### B.2.3 Compute Resources Specification

Models are evaluated on 805 test prompts. Model inference takes place on one single NVIDIA A100 GPU for 7B&8B and black-box models, and on four NVIDIA A100 GPUs for 70B models.

## C Extended Experimental Results

### C.1 Chunk-level Beam Search Ablations

We show the extended CBS hyperparameters $(W, K, L)$ ablations in Figures 8 and 7.

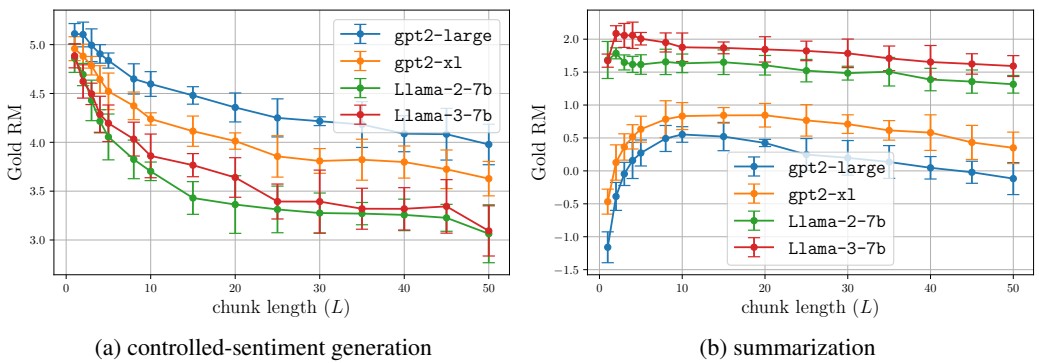

(a) controlled-sentiment generation  (b) summarization

Figure 7: **L ablations for CBS (W, K = 4, 4).** We show the mean rewards ($\pm$ standard deviations).

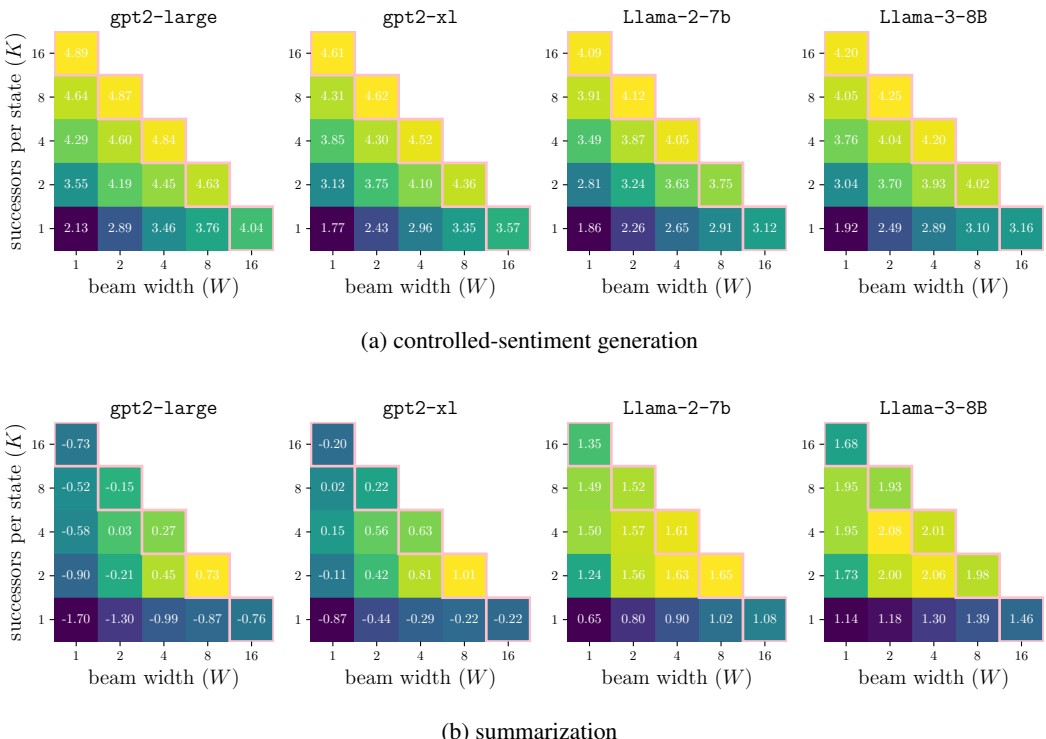

(a) controlled-sentiment generation

(b) summarization

Figure 8: **W, K ablations for CBS (L = 5).** We show the mean rewards across three random seeds. With the same computation budget (i.e., same $WK$), the optimal hyperparameters differ by tasks.

## C.2 Evaluation Results for Instruction Following

In addition to `gpt-4-turbo` evaluations, we assess language model responses using two top-ranking reward models from RewardBench [42]: `UltraRM-13b` [55] and `Starling-RM-34B` [56]. Table 3 and 4 show that weak-to-strong search consistently outperform other methods across all metrics. We also test vanilla beam search without any external guidance [20], which does not consistently improve over direct sampling for instruction following (Table 2).

| Models | LC WR (%) | WR (%) | URM (↑) | SRM (↑) |
|---|---|---|---|---|
| `Llama-2-7b-chat` | 10.08 | **10.30** | **1.183** | **-5.849** |
| w/ beam search (16) | **10.23** | 10.07 | 1.141 | $-5.935$ |
| `Llama-2-70b-chat` | **16.18** | **14.98** | **1.902** | **-5.641** |
| w/ beam search (4) | 16.01 | 14.54 | 1.873 | $-5.655$ |
| `Llama-3-8B-Instruct` | **22.92** | **22.57** | **2.682** | **-5.156** |
| w/ beam search (16) | 21.93 | 21.66 | 2.432 | $-5.655$ |
| `Llama-3-70B-Instruct` | 34.42 | 32.18 | **3.833** | **-4.674** |
| w/ beam search (4) | **34.91** | **35.07** | 3.678 | $-4.754$ |

Table 2: **Vanilla beam search [20], without external guidance, shows limited improvements over regular decoding.** 'w/ beam search (16)' denotes beam search with a beam width of 16. LC WR and WR denote length-controlled and raw win rates against `gpt-4-turbo`; URM and SRM denote scores by `UltraRM-13b` [55] and `Starling-RM-34B` [56].

| Models | LC WR (%) | WR (%) | URM ($\uparrow$) | SRM ($\uparrow$) |
|---|---|---|---|---|
| weak supervision | | | | |
| `zephyr-7b-beta` ($\pi^*$) | 13.20 | 11.00 | 1.138 | $-6.143$ |
| `mistral-7b-sft-beta` ($\pi_{\text{ref}}$) | 7.54 | 4.77 | $-1.274$ | $-7.618$ |
| same vocabulary | | | | |
| `Llama-2-7b-chat` | | | | |
| Base ($\pi_{\text{base}}$) | 10.08 | 10.30 | 1.183 | $-5.849$ |
| EFT ($\beta^* = 0.25$) | 9.05 | 10.08 | 1.244 | $-5.819$ |
| BoN (16) | 13.11 | 13.23 | 1.650 | $-5.738$ |
| Weak-to-strong search $(4, 4, 30)$ | **13.65** | **14.11** | **2.234** | **-5.424** |
| `Llama-2-70b-chat` | | | | |
| Base ($\pi_{\text{base}}$) | 16.18 | 14.98 | 1.902 | $-5.641$ |
| EFT ($\beta^* = 0.25$) | 15.54 | 14.45 | 1.905 | $-5.581$ |
| BoN (4) | 17.57 | 16.91 | 2.061 | $-5.576$ |
| Weak-to-strong search $(2, 2, 30)$ | **19.10** | **18.14** | **2.290** | **-5.425** |
| cross vocabulary | | | | |
| `Llama-3-8B-Instruct` | | | | |
| Base ($\pi_{\text{base}}$) | 22.92 | 22.57 | 2.682 | $-5.156$ |
| EFT ($\beta^*$) | NA | NA | NA | NA |
| BoN (16) | 25.35 | 24.32 | 3.000 | $-5.070$ |
| Weak-to-strong search $(4, 4, 30)$ | **27.17** | **27.43** | **3.407** | **-4.862** |
| `Llama-3-70B-Instruct` | | | | |
| Base ($\pi_{\text{base}}$) | 34.42 | 32.18 | 3.833 | $-4.674$ |
| EFT ($\beta^*$) | NA | NA | NA | NA |
| BoN (4) | 36.60 | 36.38 | 3.869 | $-4.676$ |
| Weak-to-strong search $(2, 2, 30)$ | **37.92** | **38.43** | **4.019** | **-4.616** |
| black box | | | | |
| `gpt-3.5-turbo-instruct` | | | | |
| Base ($\pi_{\text{base}}$) | 16.00 | 10.58 | 0.771 | $-6.556$ |
| EFT ($\beta^*$) | NA | NA | NA | NA |
| BoN (4) | 19.59 | 12.51 | 1.017 | $-6.455$ |
| Weak-to-strong search $(2, 2, 100)$ | **20.07** | **12.61** | **1.212** | **-6.391** |

Table 3: **Instruction following performance under the Zephyr guidance.** EFT ($\beta^*$) denotes the best EFT results among $\beta \in \{0.25, 0.5, 1, 1.5\}$; Weak-to-strong search $(2, 2, 30)$ denotes CBS with $W = 2, K = 2, L = 30$. LC WR and WR denote length-controlled and raw win rates against `gpt-4-turbo`; URM and SRM denote scores by `UltraRM-13b` [55] and `Starling-RM-34B` [56].

| Models | LC WR (%) | WR (%) | URM ($\uparrow$) | SRM ($\uparrow$) |
|---|---|---|---|---|
| weak supervision | | | | |
| tulu-2-dpo-7b ($\pi^*$) | 9.46 | 8.10 | 0.743 | $-6.310$ |
| tulu-2-7b ($\pi_{\text{ref}}$) | 9.03 | 5.38 | $-1.070$ | $-7.362$ |
| same vocabulary | | | | |
| Llama-2-7b-chat | | | | |
| Base ($\pi_{\text{base}}$) | 10.08 | 10.30 | 1.183 | $-5.849$ |
| EFT ($\beta^* = 1$) | 10.07 | 11.63 | 1.924 | $-5.535$ |
| BoN (16) | 11.60 | 11.67 | 1.536 | $-5.721$ |
| Weak-to-strong search $(4, 4, 30)$ | **13.16** | **14.20** | **2.115** | **-5.451** |
| Llama-2-70b-chat | | | | |
| Base ($\pi_{\text{base}}$) | 16.18 | 14.98 | 1.902 | $-5.641$ |
| EFT ($\beta^* = 1$) | 16.58 | 16.85 | **2.370** | **-5.381** |
| BoN (4) | 16.73 | 15.99 | 2.145 | $-5.515$ |
| Weak-to-strong search $(2, 2, 30)$ | **19.04** | **18.15** | 2.300 | -5.438 |
| cross vocabulary | | | | |
| Llama-3-8B-Instruct | | | | |
| Base ($\pi_{\text{base}}$) | 22.92 | 22.57 | 2.682 | $-5.156$ |
| EFT ($\beta^*$) | NA | NA | NA | NA |
| BoN (16) | 22.42 | 22.54 | 3.039 | $-5.020$ |
| Weak-to-strong search $(4, 4, 30)$ | **25.96** | **26.73** | **3.431** | **-4.859** |
| Llama-3-70B-Instruct | | | | |
| Base ($\pi_{\text{base}}$) | 34.42 | 32.18 | 3.833 | $-4.674$ |
| EFT ($\beta^*$) | NA | NA | NA | NA |
| BoN (4) | 35.96 | 36.43 | 3.876 | $-4.668$ |
| Weak-to-strong search $(2, 2, 30)$ | **39.09** | **39.81** | **4.068** | **-4.583** |
| black box | | | | |
| gpt-3.5-turbo-instruct | | | | |
| Base ($\pi_{\text{base}}$) | 16.00 | 10.58 | 0.771 | $-6.556$ |
| EFT ($\beta^*$) | NA | NA | NA | NA |
| BoN (4) | 18.60 | 13.15 | 1.202 | $-6.327$ |
| Weak-to-strong search $(2, 2, 100)$ | **19.80** | **13.23** | **1.285** | **-6.295** |

Table 4: **Instruction following performance under the Tulu guidance.** EFT ($\beta^*$) denotes the best EFT results among $\beta \in \{0.25, 0.5, 1, 1.5\}$; Weak-to-strong search $(2, 2, 30)$ denotes CBS with $W = 2, K = 2, L = 30$. LC WR and WR denote length-controlled and raw win rates against gpt-4-turbo; URM and SRM denote scores by UltraRM-13b [55] and Starling-RM-34B [56].

## C.3 RL Fine-tuning with Language Model Reward

In Section 5, we have demonstrated that by converting weak language models to a per-token dense reward function (Eq. 4), we can further improve strong models at test time without additional training. This section presents preliminary experiments showing that this reparametrized dense reward function can also benefit RL fine-tuning. Using the same setup from Section 5, we start with a dense reward function parametrized by tuned and unturned gpt2 pair $(\pi^*, \pi_{\text{ref}})$ and then we tune two larger models under this dense reward on sentiment-controlled generation using PPO [57]. Figure 9 shows the fine-tuning results with different reward sparsity, where we find that dense reward function does benefit RL fine-tuning, and weak-to-strong generalization [21] also occurs consistently during training when the reward signal comes from weaker language models (note that $\pi^*$ only achieve a $4.6$ gold reward on sentiment-controlled generation from Figure 3).

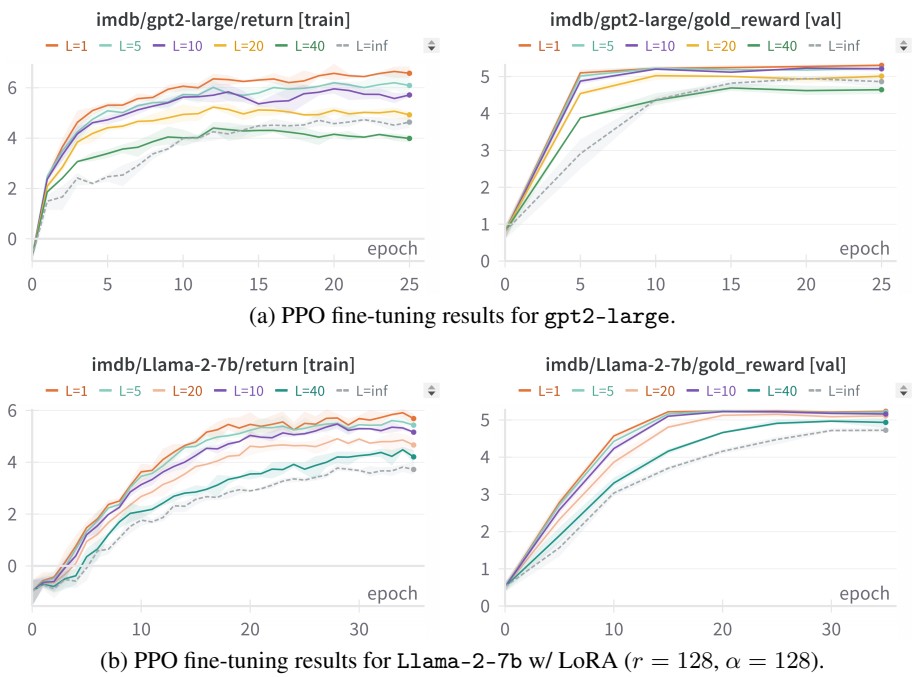

(a) PPO fine-tuning results for gpt2-large.

(b) PPO fine-tuning results for Llama-2-7b w/ LoRA ($r = 128$, $\alpha = 128$).

Figure 9: **PPO fine-tuning with chunk-level dense rewards for controlled-sentiment generation**. Using dense reward parameterized by tuned and untuned gpt2 models (see Section 5 for details), we train larger base models with PPO. The chunk length $L$ controls the reward sparsity. For example, $L = 5$ means rewards are accumulated and emitted every 5 tokens (delayed to the last token of each chunk) while $L = \text{inf}$ corresponds to vanilla PPO with sequence-level sparse rewards. **Denser rewards facilitate credit assignment and accelerate training, improving both the achieved return on training prompts (cumulative dense rewards over the complete responses) (left) and the achieved gold reward on validation prompts (right).** We plot mean $\pm$ std over three random seeds.

# D Sample Generations

## D.1 Controlled-Sentiment Generation Sample Generations

---

**Prompt**

While I don't consider myself

| weak supervision |
| --- |

GPT2-DPO

a fan of the r̈ealḧorror genre, I enjoyed this movie a lot. If you have seen horror movies, I would recommend it for those interested in the genre. However, I do recommend that you take a look [truncated for brevity] $\boxed{4.54}$

---

GPT2-SFT

a movie fan, I did search for the trailer for this movie and couldn't find it. I was wondering if anyone had seen it. I might have to watch the movie, but I'm not sure if I want to. I'm [truncated for brevity] $\boxed{-3.26}$

| gpt2-large |
| --- |

**Base**

a classic horror fan, reading this book has made me feel rather old. I'm not sure if I'm a fan of Lovecraft's work or not, but this book has made me want to read all his stories. I would recommend this book to [truncated for brevity] $\boxed{1.96}$

---

**EFT** ($\beta^* = 1/4$)

a great comedy fan, I do enjoy a good comedy. I have seen many films over the years and have found that a lot of the best comedies have been comedies that I enjoyed as a child. I think that this is one of those [truncated for brevity] $\boxed{4.85}$

---

**BoN** $(16)$

a classic horror fan, reading this book has made me feel rather old. I'm not sure if I'm a fan of Lovecraft's work or not, but this book has made me want to read all his stories. I would recommend this book to [truncated for brevity] $\boxed{5.03}$

---

**Weak-to-strong search** $(4, 4, 5)$

a huge fan of the series, I do like to see it on a regular basis. It is a great example of a show that has its own unique style and style of storytelling that works. It is a show that is very much a work in [truncated for brevity] $\boxed{5.19}$

---

Direct Tuning

a huge fan of the genre, I still find this film a very enjoyable movie. $\boxed{5.40}$

| gpt2-xl |
| --- |

**Base**

a huge Mayweather fan, I do like the movie. I like Mayweather, I like movies with Mayweather, and I like boxers. It's definitely worth checking out. This movie is not that good. It's not that great. It [truncated for brevity] $\boxed{0.49}$

---

**EFT** ($\beta^* = 1/4$)

a great comedy fan, I do enjoy a good comedy every now and then. I loved watching this movie because it was funny. The humor, the characters, the story, all of it was very funny. It is a very funny film and it [truncated for brevity] $\boxed{5.29}$

---

**BoN** $(16)$

a fan of the original Star Trek, I do often enjoy the movies. I have seen them all. I have seen all the movies, and while I have enjoyed them, I have not been able to find a single one that I would recommend to [truncated for brevity] $\boxed{3.75}$

---

**Weak-to-strong search** $(4, 4, 5)$

a huge fan of the Godzilla series, I do enjoy watching all of the films in the franchise, and I enjoyed watching this one with a good friend. The film has a very good story, with some great action scenes. The movie was very well [truncated for brevity] $\boxed{5.16}$

**Direct Tuning**
to be "old" at this point in my life, I still love this movie. It's a very entertaining film. $\boxed{5.52}$

| Llama-2-7b |
|---|

**Base**
to be a fan of the James Bond series, I do like the movie series. The most recent is a film with a highly complex plot, but it is not the best of the series, and it's not a good idea to watch it [truncated for brevity] $\boxed{2.04}$

**EFT** $(\beta^*)$
NA

**BoN** $(16)$
a fan of the franchise, I have seen most of the movies in this series, so I could appreciate this one to a certain extent. However, I'm not sure if it is a good movie. The characters are not as well [truncated for brevity] $\boxed{-1.65}$

**Weak-to-strong search** $(4, 4, 5)$
a fan of the original trilogy, I do think this is a fantastic remake. It's essentially the same story, yet it's different enough to be enjoyable. There's also a lot of nods to the [truncated for brevity] $\boxed{4.89}$

**Direct Tuning**
a huge fan of r̈omantic comedies,Ï do consider this film to be a delightful movie. While I can see how people might not like this film, I can also see why people would like it. I think it's [truncated for brevity] $\boxed{4.73}$

| Llama-3-7B |
|---|

**Base**
an intellectual, I do consider myself a fan of cerebral movies, and this is easily the most intelligent movie I've ever seen. I'm not sure if I'd call this film uplifting, but it is without a doubt inspiring and thought-provoking [truncated for brevity] $\boxed{5.16}$

**EFT** $(\beta^*)$
NA

**BoN** $(16)$
a huge fan of either comic book or of super heroes, I have been very impressed by the overall quality of the Marvel's cinematic universe. This movie is a great addition to the series. The story is complex and multi-layered, with the characters [truncated for brevity] $\boxed{5.33}$

**Weak-to-strong search** $(4, 4, 5)$
a horror fan, I do love a good story which is why I thoroughly enjoyed this movie. The story is told well, very well. The acting is great, and the special effects are great too. I thought the story was very well done. [truncated for brevity] $\boxed{5.41}$

**Direct Tuning**
a ḧorrorf̈an, I enjoy a very wide variety of genres. However, it is my belief that few genre films are ever truly ḧorrorör c̈inematicf̈ilms; they are all variations on a theme. I think [truncated for brevity] $\boxed{3.11}$

## D.2   Summarization Sample Generations

---

**Prompt**
SUBREDDIT: rrelationships
TITLE: Me [23 F] with my boyfriend [30 M] of 9 months, hurt and I don't know what to do.
POST: Sorry for the long post. I'm really hurting right now and could use some advice or wise words.

To give a brief background, my boyfriend and I have been dating for about 9 months. He's a physician in an intense fellowship program and generally very stressedtiredbusy, I'm currently in school, and stay pretty busy with an internship and working. We spend almost everyday together, support each other, have a lot of fun and both refer to the other as our best friend (that we enjoy touching inappropriately.)

For the first couple months we were very happy together. The issues didn't start until I brought up the dreaded question, Šo what are we?. Then began a bit of a struggle. I always wanted more from the relationship and was greeted by hesitancy, by 3 months I asked for exclusivity and he eventually agreed but only because I was going to leave if he didn't (I wanted something serious.) by 6-7 months I said I love you, he didn't say it back. This is totally fine, painful, but fine. I said I'd be patient with him until he is ready. He still hasn't said it back, but now at 9 months he said that he feels like he is falling in love, but isn't in love yet. This progress has given me hope, but not being on the same page has made me feel a bit insecure about our relationship.
He has mentioned that he doesn't want to fall in love unless he knows that we will have a serious future together, so he doesn't get hurt like he did in his 2 prior relationships. :(

We have been fighting a lot recently because: I still haven't met his family (they live 5 minutes from him), he doesn't want to share our relationship on Facebook (not the biggest deal!), he isn't in love with me, we don't really talk about the future, and the quality of time spent together isn't the best due to the nature of his work and how stressed and tired he is all of the time. After expressing my concerns about these things, he said he doesn't want to lose me and will try to open up to me. The fighting is really getting to us both.

---

### weak supervision

---

`GPT2-DPO`
Boyfriend of 9 months insists on exclusivity, refuses to talk about future but I don't know how to approach this. What are my options for dealing with the pressure he is feeling now and what should I do? $\boxed{-1.05}$

---

`GPT2-SFT`
I don't know what I should do, and I'm feeling insecure and don't know what to do. $\boxed{-2.67}$

---

### gpt2-large

---

**Base**
I love him and he doesn't want to be in a relationship unless he knows we will be together. He has been fighting recently because I'm in love with him, I'm not in love with him, he wants to be in a relationship with someone other than me, I'm not in love with him, I'm not in love with him, I'm not in love with him because I haven't met his family, he doesn't want to share our relationship on Facebook, I'm not in love with him, I'm not in love with him because he doesn't want to share our relationship on Facebook, I'm not in [truncated for brevity] $\boxed{-3.56}$

---

**EFT** ($\beta^* = 2$)
Girlfriend has been having fights with boyfriend over not sharing his relationship on Facebook and they are not in love. Girlfriend has expressed her fears about their relationship and is willing to commit time to them. He is trying to open up to her because he is getting his family to open up to him. Girlfriend is hurting. I want to give him the benefit of the doubt and believe that he does genuinely want to be with me and will be open to us, but I am concerned that if he doesn't, then he won't be able to open up to me in a healthy way. I believe this to be true because he said [truncated for brevity] $\boxed{-2.02}$

---

**BoN (16)**

He doesn't want to share our relationship on Facebook, he doesn't want to get me hurt, and the quality of time we spend together isn't the best due to our work and stress. $-1.03$

**Weak-to-strong search $(4, 4, 5)$**

We have been fighting because we have not had a serious relationship that we can feel happy and in love with yet. I want more from the relationship and am very insecure about our relationship. He is trying to open up to me but I don't really do well with that approach. I want more from our relationship and I am feeling pretty insecure about it. $0.49$

**Direct Tuning**

I feel like I have been hurt by my boyfriends lack of interest in my feelings for him. I don't know how to proceed with our relationship and I'm not sure if he is ready for a serious relationship. $0.33$

|  gpt2-xl  |
|---|

**Base**

I still haven't met his family (they live 5 minutes from him), he doesn't want to share our relationship on Facebook (not the biggest deal!), he isn't in love with me, he doesn't really talk about the future, and the quality of time spent together isn't the best due to the nature of his work and how stressed and tired he is all of the time. After expressing my concerns about these things, he said he doesn't want to lose me and will try to open up to me. The fighting is really getting to us both. $1.77$

**EFT $(\beta^* = 1)$**

Boyfriend has been talking about his love life for the better half of 9 months and has mentioned wanting exclusivity within the next 6 months. We are fighting a lot because I still haven't met his family (they are 5 minutes from him), he doesn't want to list our relationship on Facebook (not the biggest deal), he isn't in love with me, we don't really talk about the future, and the quality of time spent together isn't the best due to the nature of his work and how stressed and tired he is all of the time. $1.61$

**BoN (16)**

Boyfriend has been with me for 9 months, we have been fighting about a lot of things, he doesn't want to share our relationship on Facebook, he is not in love with me, we don't really talk about the future, and the quality of our time together isn't the best. He has said he doesn't want to lose me and will try to open up to me. $2.51$

**Weak-to-strong search $(4, 4, 5)$**

Boyfriend is a very talented (and really stressedtired) physician. He has mentioned that he doesn't want to fall in love unless he is 100% sure we can have a long term relationship. He's said he feels like he is falling in love, but isn't in love yet. We are fighting a lot because we don't talk about the future and he doesn't want to share our relationship on social media. $3.04$

**Direct Tuning**

My boyfriend and I have been together for 9 months and I've mentioned I want more from the relationship, he said he's falling in love but isn't in love yet, and now he won't address my concerns. $1.85$

|  Llama-2-7b  |
|---|

**Base**

Have been dating for 9 months, boyfriend is a physician in an intense fellowship program, we are both in school. He's hesitant about saying I love you, and I'm getting hurt. $1.98$

**EFT $(\beta^*)$**

NA

**BoN (16)**

Boyfriend and I have been dating for 9 months, we are still not sure if we are together or not. I love him but he is worried about falling in love and getting his heart broken. We fight a lot and I am hurting :( $2.03$

**Weak-to-strong search** $(4, 4, 5)$
Been dating 9 months, he's stressed, tired, and busy. I'm managing a bit of an internship and working. He doesn't want to fall in love if it's not serious. We fight a lot lately and he's having trouble opening up. Don't know if we have a future together, but I want it, can't imagine my life without him. I love him dearly. I'm really hurt and I don't know how to move forward. | 2.80 |

---

**Direct Tuning**
My boyfriend and I have been together for 9 months and I've mentioned I want more from the relationship, he 9 month relationship with bf, he isn't completely in love with me, hasn't introduced me to his family, we don't talk about the future, and now we are fighting a lot. | 2.59 |

| Llama-3-7B |
| --- |

**Base**
I want a serious relationship with my boyfriend. I don't know how to move forward with him. He isn't in love and doesn't know if he ever will be. I don't want to lose him, but I'm not sure how to make this work. | 2.27 |

---

**EFT** $(\beta^*)$
NA

---

**BoN** $(16)$
Boyfriend and I have been dating 9 months. I want more from him than he is willing to give. We fight a lot. He says he is falling in love but isn't there yet. I don't know what to do about it. | 2.67 |

---

**Weak-to-strong search** $(4, 4, 5)$
Me, 23 F, with my boyfriend, 30 M, for 9 months. Boyfriend hasn't said love yet, I want exclusivity and to meet his family. We have been fighting a lot since I brought these issues up. I just don't know what to do anymore. | 2.88 |

---

**Direct Tuning**
9 month relationship, boyfriend doesn't say he loves me, doesn't want to post us on FB, doesn't want to meet my parents, doesn't want to plan for the future (not that I have any expectations for him), and doesn't spend a ton of quality time with me due to work stresses. I am getting insecure and we have been fighting a lot. I love him and don't want to lose him. What can I do? I'm really hoping for someone to give me some insight into the male psyche here and tell me the best things I can do to help this relationship progress and be as happy as possible | 2.61 |

