# OpenReview forum: "Weak-to-Strong Search: Align Large Language Models via Searching over Small Language Models"
_NeurIPS.cc/2024/Conference — NeurIPS 2024 poster_

### Official Review · Reviewer_Xaca · 2024-07-09

**Soundness:** 3
**Presentation:** 3
**Contribution:** 3
**Rating:** 7
**Confidence:** 2

**Summary:**

In this paper, they frame alignment as a decoding time problem, allowing the large language model to be frozen. They do this by parametrizing a reward function with the difference in the log-likelihood between small untuned and tuned language models. Interestingly, their approach does not require shared vocabulary and is applicable to black-box LLMs. They perform experiments on controlled-sentiment generation, summarization, and instruction following.

**Strengths:**

- The paper is well written and organized.
- The motivation is clear from the beginning.
- The experimental setting is adequate and the results are good.

**Weaknesses:**

Although there’s a related work section, I believe the comparison between the proposed method and other methods, e.g., the ones that rely on small language models to aid the alignment of LLMs, should be described in more detail. In particular, the novelty of the proposed method should be made more evident; for non-experts, I think it’s a bit difficult to understand if some parts are novel or if they already exist in previous work. I also think that some citations might be missing (see my questions below).

**Questions:**

Questions:
- Is it \pi_base in L181?
- What about the vocabulary of the small tuned and tuned models? Does it need to be shared?
- How does your work compare with that of Li et al. (ACL 2023)? What about Zhao et al. (2024)?

Minor comments:
- BoN is defined in L204 but it’s used before (e.g., L166).

References:

- Contrastive Decoding: Open-ended Text Generation as Optimization (Li et al., ACL 2023)
- Weak-to-Strong Jailbreaking on Large Language Models (Zhao et al., 2024)

**Limitations:**

Yes.

---

> ### Author Rebuttal · Authors · 2024-08-05
>
> Thank you for your valuable comments. To address your questions:
>
> ---
>
> > Ｗ1: Although there’s a related work section, I believe the comparison between the proposed method and other methods, e.g., the ones that rely on small language models to aid the alignment of LLMs, should be described in more detail. In particular, the novelty of the proposed method should be made more evident; for non-experts, I think it’s a bit difficult to understand if some parts are novel or if they already exist in previous work. I also think that some citations might be missing (see my questions below).
>
> Thank you for your suggestions on more detailed related works. Please see our response to Q3. We will incorporate these discussions into the camera-ready version.
>
> ---
>
> > Q1: Is it \pi_base in L181?
>
> Thank you for pointing out this typo. We will fix it in the camera-ready version.
>
> ---
>
> > Q2: What about the vocabulary of the small tuned and tuned models? Does it need to be shared?
>
> In principle, our method does not require the vocabularies of small tuned and untuned models to be shared because we calculate the log-likelihood difference of small models over natural language chunks (which can be tokenized differently according to each model's vocabulary) rather than over particular tokens. In practice, though, when the former is fine-tuned from the latter, they almost always share the same vocabulary. Exploring cross-vocabulary guidance from different small model families (e.g., Llama2 and Llama3) is an interesting direction we plan to pursue in future work.
>
> ---
>
> > Q3: How does your work compare with that of Li et al. (ACL 2023)? What about Zhao et al. (2024)?
>
> Thank you for highlighting these related works. They are indeed quite relevant to ours, and we will discuss them further in our camera-ready paper. Both [1] and [2] can be interpreted as special cases of emulated fine-tuning (EFT) [3], which is a key baseline for our experiments. EFT adjusts a target model's output token distribution based on other models' distributions. Specifically, [2] uses EFT to remove a model's safety guardrails, while [1] applies EFT where $\pi_{\text{base}}$ and $\pi^*$ are identical (see L210 for EFT formulations). However, these token-level manipulations have drawbacks: (1) they require all models to share the same vocabulary, (2) they do not apply to black-box models without full transparency of their output distributions, and (3) they are empirically less effective than **our proposed test-time search method, which is the only method that achieves weak-to-strong generalization, enhancing strong models with weak guidance**.
>
> ---
>
> > C1: BoN is defined in L204 but it’s used before (e.g., L166).
>
> Thank you for pointing out this issue. We will fix it in the camera-ready version.
>
> ---
>
> If these answers do not fully address your concern, we are more than willing to offer additional clarifications.
>
>
> [1] Li, Xiang Lisa, et al. "Contrastive decoding: Open-ended text generation as optimization." arXiv preprint arXiv:2210.15097 (2022).\
> [2] Zhao, Xuandong, et al. "Weak-to-strong jailbreaking on large language models." arXiv preprint arXiv:2401.17256 (2024).\
> [3] Mitchell, Eric, et al. "An emulator for fine-tuning large language models using small language models." arXiv preprint arXiv:2310.12962 (2023).

---

> > ### Comment · Reviewer_Xaca · 2024-08-08
> >
> > Thank you for answering my questions. I have read the rebuttal and the other reviews. I believe that incorporating some of this discussion in the updated version of the paper is a good idea.

---

> > > ### Author Response · Authors · 2024-08-08
> > >
> > > Thanks for your response and feedback!

---

### Official Review · Reviewer_2D8T · 2024-07-16

**Soundness:** 3
**Presentation:** 4
**Contribution:** 3
**Rating:** 6
**Confidence:** 4

**Summary:**

The paper introduces the "weak-to-strong search" method for aligning a stronger large language models (LLMs) by leveraging two weaker LLMs during test-time without requiring fine-tuning of the large models. This approach aims to improve alignment by maximizing the log-likelihood difference between tuned and untuned small models while using the frozen large model for decoding. The method demonstrates effectiveness in various tasks, including sentiment-controlled generation, summarization, and instruction following.

I generally think this is an inspiring paper towards acceptance. The reason I rate it a weak accept rather than accept is I personally think an ablation study of using the log-likelihood difference to conduct token/chunk-wise PPO is necessary. It can make the paper more complete but lacking it is kinda fine though.

**Strengths:**

1. The weak-to-strong search method is novel and provides a computationally efficient way to align LLMs without fine-tuning.
2. The method is shown to work across different tasks and with both white-box and black-box models.
3. The method is well-grounded in theory, with detailed mathematical formulations and explanations.

I really like the overall presentation of this paper. The reading flow is good; better than many other ones in this NeruIPS review cycle. :D

**Weaknesses:**

1. Obtaining a tuned weaker model is still not a very trivial thing, and the performance of $\pi_\text{base}$ on downstream tasks could heavily depend on the tuned weaker model as well. There is not too much discussion regarding it.
2. As we can formulate a partial reward function using a tuned/untuned model pairs, apart from using it to guide decoding, a natural next step/ablation study is using it to conduct PPO. It is worth trying this experiment and compare the downstream performance as well as the overall cost.

**Questions:**

N/A no questions here; the paper is presented very clearly.

**Limitations:**

The paper includes a limitation section at the end.

---

> ### Author Rebuttal · Authors · 2024-08-05
>
> Thank you for your valuable comments and we're glad you enjoyed the paper. To address your questions:
>
> ---
>
> > W1: Obtaining a tuned weaker model is still not a very trivial thing, and the performance of $\pi_{\text{base}}$ on downstream tasks could heavily depend on the tuned weaker model as well. There is not too much discussion regarding it.
>
> There are two practical reasons why we do not obtain a collection of weak models $\pi^*$ to analyze how their performance influences weak-to-strong search:
>
> 1. When tuning the weak model $\pi^*$ ourselves (e.g., controlled-sentiment generation and summarization experiments): In the context of alignment, reward models are usually trained on relative preferences and it is standard practice to use the reward model with the highest validation accuracy [1,2]. Since the tuned weaker models are eventually used as rewards in the downstream search, we follow the standard practice to use the $\pi^*$ checkpoint with the highest preference classification accuracy (we use DPO fine-tuning throughout the paper) and do not perform additional ablations.
>
> 2. When reusing off-the-shelf tuned weak model $\pi^*$ (e.g., instruction-following experiments): We have little control over these models' performance as their weights are fixed, and there are not enough open-source weak models to form a spectrum for ablation.
>
> However, we have demonstrated that weak-to-strong search works consistently across various weak models and outperforms other baselines.
>
> ---
>
> > W2: As we can formulate a partial reward function using a tuned/untuned model pairs, apart from using it to guide decoding, a natural next step/ablation study is using it to conduct PPO. It is worth trying this experiment and compare the downstream performance as well as the overall cost.
>
> Thank you for the valuable suggestion. Exploring if dense rewards can benefit PPO is indeed interesting. **Figure 2 in the supplementary PDF (global response) shows that dense rewards do accelerate PPO training. Weak-to-strong search (ours) outperforms vanilla PPO with sequence-level reward by a large margin, but it is less effective than chunk-level PPO.**
>
> **Despite these promising ablation results, we need to kindly emphasize that this work focuses more on training-free alignment and a framework for weak-to-strong generalization.** The key advantage of training-free method is its ability to guide a large model to approximate fine-tuning without significant computational resources. While training-based methods are promising, they are resource-intensive and unstable. For instance, we do not have the resources to fine-tune a 70B model, but we can approximate fine-tuning results using weak-to-strong search.
>
> We appreciate the suggestions on experimenting with chunk-level PPO. **These ablations do make our paper more complete and we will include these results in the camera-ready version. However, a more thorough exploration and analysis of chunk-level PPO likely deserves another paper and will be addressed in future work.** For instance, given that dense rewards implicitly handle credit assignment, do we need value function modeling and GAE in PPO to stabilize training? Could vanilla REINFORCE achieve similar results as PPO under such dense rewards? We leave these questions for future investigation.
>
> ---
>
> If these answers do not fully address your concern, we are more than willing to offer additional clarifications.
>
>
> [1] Ouyang, Long, et al. "Training language models to follow instructions with human feedback." Advances in neural information processing systems 35 (2022): 27730-27744.\
> [2] Rafailov, Rafael, et al. "Direct preference optimization: Your language model is secretly a reward model." Advances in Neural Information Processing Systems 36 (2024).

---

> > ### Comment · Reviewer_2D8T · 2024-08-07
> >
> > Thanks for your response and additional response! It currently looks good to me.
> >
> > I'll follow up if there are further questions. So far so good.

---

> > > ### Author Response · Authors · 2024-08-08
> > >
> > > Thank you for your prompt response! And I wanted to kindly remind you that you rated our paper as "weak accept" due to some of the weaknesses you mentioned. If we have satisfactorily addressed these weaknesses in our rebuttal, would you be open to increasing your score? If not, are there any other clarifications I can provide to assist in your evaluation?

---

> > > > ### Comment · Reviewer_2D8T · 2024-08-08
> > > >
> > > > It's too early to talk about the rating. I'll reach out if I need further clarifications. I might recalibrate my rating during the AC-reviewer discussion period but it depends on the discussion and other reviewers as well.

---

> > > > > ### Author Response · Authors · 2024-08-08
> > > > >
> > > > > I totally get that it's too early to finalize the rating. Thanks again for your thoughtful comments and suggestions!

---

### Official Review · Reviewer_mz4P · 2024-07-17

**Soundness:** 3
**Presentation:** 3
**Contribution:** 3
**Rating:** 6
**Confidence:** 4

**Summary:**

The paper proposes a search/decoding method "weak-to-strong search" for improving LLM's performance at test time by leveraging the log likelihood from small language models. Specifically, it utilizes the log likelihood differences between tuned and untuned small models to guide the decoding of larger models. This approach avoids the computationally intensive process of fine-tuning large models by using smaller, more manageable models for alignment at inference time, guiding the large model to generate better responses. The paper demonstrates the effectiveness of this method across various tasks, including controlled-sentiment generation, summarization, and instruction-following tasks.

**Strengths:**

* The paper introduces a novel approach to improve large model alignment at inference time, which reduces computational costs compared to fine-tuning methods. The idea of leveraging log likelihood differences between untuned and tuned small language models is quite interesting.
* The method is tested across multiple tasks and has shown effectiveness. Results demonstrate improvements in multiple tasks.
* Ablation studies offer further analysis and understanding of the hyperparameters and components of the method.

**Weaknesses:**

* Although the method avoids the computational costs associated with fine-tuning, the paper does not explain clearly the overhead introduced during inference. How does weak-to-strong search impact memory usage and inference speed?
* How does the proposed method compare to other inference-time techniques, such as in-context learning (using few-shot examples) and prompting methods like CoT?
* The method appears to be broadly applicable and not strictly limited to a weak-to-strong approach. For smaller models, such as Llama3-8B as used in the experiments, why would this method be preferred over techniques like LoRA fine-tuning, which can be performed at a low cost and yield better performance? Additionally, if the target model has already been fine-tuned, can the proposed method still enhance its performance?
* (minor) It would be interesting to explore how weak-to-strong search could enhance a model's reasoning abilities, for example on math datasets like GSM8K.

**Questions:**

See weaknesses above.

**Limitations:**

Yes, the authors discussed limitations in the paper.

---

> ### Author Rebuttal · Authors · 2024-08-05
>
> Thank you for your valuable comments. To address your questions:
>
> ---
>
> > W1: The paper does not explain clearly the overhead introduced during inference. How does weak-to-strong search impact memory usage and inference speed?
>
> Throughout the paper, we select the hyperparameters for CBS (W, K, L) to ensure its inference costs are comparable to BoN sampling. Specifically, CBS samples W\*K trajectories in parallel from the large base (target) model, while BoN samples N trajectories, so we always ensure W\*K=N. The only overhead of CBS over BoN is running small models chunk-wise for intermediate guidance/evaluation, while BoN runs them once at the end. These overheads are negligible when the chunk length is large or the small models are much smaller than the base model, which is often the case. **Figure 1 in the supplementary PDF (global response) empirically supports this by comparing the GPU memory usage and inference speed of different test-time methods for instruction-following.** The GPU memory usage and inference speed of CBS with our chosen hyperparameters are close to BoN sampling, while CBS is substantially more effective in steering language models and achieves weak-to-strong generalization (which BoN fails to do). We will include these analyses in our camera-ready version.
>
> ---
>
> > W2: How does the proposed method compare to other inference-time techniques, such as in-context learning (using few-shot examples) and prompting methods like CoT?
>
> For summarization, the base (target) models are already few-shot prompted (see Section B.1.6). For instruction-following, the base (target) models are chat models (see L264-266) using a chat template, which does not naturally support few-shot exemplars. Therefore, we test few-shot prompting on the controlled-sentiment generation task, and the results are shown below. **These results show that our method is more effective than few-shot prompting and, more importantly, complementary to it (they can be combined to achieve the best results). For new empirical results on other test-time baselines, please refer to our response to Reviewer ct59's W1.**
>
> |                                | gpt2-large | gpt2-xl | Llama-2-7b | Llama-3-8B |
> |--------------------------------|:----------:|:-------:|:----------:|:----------:|
> | Base                           |    2.127   |  1.711  |    1.857   |    1.915   |
> | Base + 3 shot                  |    2.670   |  2.507  |    2.847   |    2.934   |
> | Weak-to-strong search  (ours)    |    *4.837*  |  *4.522*  |    *4.055*   |    *4.195*   |
> | Weak-to-strong search  (ours) + 3 shot  |    **5.028**   |  **4.855**  |    **4.448**   |    **4.647**   |
>
> ---
>
> > W3.1: For smaller models, such as Llama3-8B as used in the experiments, why would this method be preferred over techniques like LoRA fine-tuning, which can be performed at a low cost and yield better performance?
>
> Our training-free framework offers three key advantages over parameter-efficient fine-tuning (e.g., LoRA):
>
> 1. **No need for accessing model weights**: Our method does not need access to the weights of the base (target) models. We only require the ability to sample from potentially black-box models and periodically select promising response states (see our black-box GPT-3.5 experiments in Figure 6).
>
> 2. **No need for fine-tuning data**: We sometimes lack the data necessary for fine-tuning. For instance, Meta fine-tunes and releases their models but does not release their fine-tuning datasets. Our method can reuse open-sourced models as guidance to approximate the results of fine-tuning target models on these proprietary datasets.
>
> 3. **Performance**: Our method can close the performance gap with full-parameter fine-tuning in controlled sentiment generation, summarization (Figure 3), and mathematical reasoning (see our response to W4).
>
> ---
>
> > W3.2: Additionally, if the target model has already been fine-tuned, can the proposed method still enhance its performance?
>
> Yes, this is exactly what we do for instruction-following experiments, where we use tuned chat models as base (target) models (see L260-266). Empirical results in Figure 6 (we apologize for the caption confusion; each model ID in Figure 6 stands for its chat version, e.g., Llama3-8B refers to $\texttt{Llama-3-8B-Instruct}$), with detailed results in Tables 3 and 4, show that our method can enhance an already tuned model with weak guidance, enabling consistent weak-to-strong generalization.
>
> ---
>
> > W4: (minor) It would be interesting to explore how weak-to-strong search could enhance a model's reasoning abilities, for example on math datasets like GSM8K.
>
> Exploring how our method enhances a model's reasoning abilities is indeed interesting. **We verify that our method demonstrates weak-to-strong generalizaition on mathematical reasoning when small models** $\pi^*$ **are specifically tuned for reasoning**. We will include these ablation results in the camera-ready version. Specifically, we reuse two small model pairs tuned for mathematical reasoning:
>
> 1. Qwen2-7B-Instruct-Step-DPO ($\pi^*$) (GSM: 88.5), Qwen2-7B-Instruct ($\pi_{\text{ref}}$) (GSM: 82.3);
> 2. DeepSeekMath-RL-7B ($\pi^*$) (GSM: 88.2), DeepSeekMath-Instruct-7B ($\pi_{\text{ref}}$) (GSM: 82.9).
>
> These models are publicly available on HuggingFace, with $\pi^*$ tuned from $\pi_{\text{ref}}$. We then use these pairs to steer a larger base (target) model already strong in mathematical reasoning: Qwen2-72B-Instruct (GSM: 91.1). Weak-to-strong search with Qwen2-7B guidance enhances the performance of the 72B untuned version from 91.1 to 94.47, while weak-to-strong search with DeepSeek-7B enhances its performance from 91.1 to 94.24. **Notably, weak-to-strong search closes the performance gap and even outperforms directly fine-tuning the large model: Qwen2-72B-Instruct-Step-DPO (GSM: 94.0).**
>
> ---
>
> If these answers do not fully address your concern, we are more than willing to offer additional clarifications.

---

> > ### Comment · Reviewer_mz4P · 2024-08-11
> > **Thank you for your rebuttal**
> >
> > Thanks for your response. I appreciate the rebuttal and have read other reviews. I maintain my score as 6 and will engage in further discussion with other reviewers and AC if needed.

---

> > > ### Author Response · Authors · 2024-08-12
> > >
> > > Thanks for your response and feedback!

---

### Official Review · Reviewer_ct59 · 2024-07-19

**Soundness:** 3
**Presentation:** 3
**Contribution:** 2
**Rating:** 6
**Confidence:** 3

**Summary:**

This paper addresses the alignment of large language models without the need for fine-tuning. It conceptualizes the alignment process as a search problem, leveraging the log-likelihood difference between small tuned and untuned language models as both a reward and a critic. By transforming a sparse preference reward into a per-token dense reward, the method achieves weak-to-strong generalization. The effectiveness of the proposed approach is empirically validated through controlled sentiment generation, summarization, and an instruction-following benchmark.

**Strengths:**

1. The paper is well-written, and the empirical results are promising.
2. The method is well-motivated and theoretically sound.
3. The proposed approach provides token-level guidance instead of sparse sequence-level rewards, enhancing precision and control.

**Weaknesses:**

1. The paper does not include comparisons with existing decoding-based alignment baselines [1,2].

2. The token-based MDP formulation is largely derived from existing work [3]. The main innovation of this paper lies in using the log-likelihood difference between two small models for guidance based on the existing formulation.


[1] ARGS: Alignment as Reward-Guided Search, ICLR 2024

[2] Controlled Decoding from Language Models, ICML 2024

[3] From r to Q*: Your Language Model is Secretly a Q-Function.

**Questions:**

What is the difference between the formulation in section 4.1 and the results in [3]?

**Limitations:**

The paper has discussed limitations.

---

> ### Author Rebuttal · Authors · 2024-08-05
>
> Thank you for your valuable comments. To address your questions:
>
> ---
>
> > W1: The paper does not include comparisons with existing decoding-based alignment baselines [1,2]
>
> We would like to clarify that a key advantage of our work is its ability to **reuse off-the-shelf language models for test-time guidance**. Given the numerous open-source language models available online for different target tasks (e.g., mistral->zephyr for chatting, llama->codellama for coding), **our approach can almost always be training-free**. In contrast, ARGS [1] requires a reward model (often proprietary), and Controlled Decoding [2] requires a value function trained from scratch, **making them not truly training-free**.
>
> **However, we agree that comparing our approach with these mentioned decoding-based alignment baselines is essential for comprehensive performance benchmarking. Therefore, we have tested ARGS [1] and Controlled Decoding (CD-fudge variant) [2] on controlled-sentiment generation and summarization, and the results (averaged over three random seeds) are shown below.** Note that vanilla ARGS [1] and Controlled Decoding [2] operate on a per-token level and are not applicable to cross-vocabulary guidance (i.e., guiding Llama-2 and Llama-3). While ARGS and Controlled Decoding do enhance the performance of base models to some extent, they are less effective than our method.
>
> |                       |    imdb    |   imdb    | summarization | summarization |
> |:--------------------- |:----------:|:---------:|:-------------:|:-------------:|
> |                       | gpt2-large |  gpt2-xl  |  gpt2-large   |    gpt2-xl    |
> | Base                  |   2.127    |   1.711   |    -1.703     |    -0.871     |
> | Weak-to-strong search (ours) | **4.837**  | **4.522** |   **0.272**   |   **0.633**   |
> | ARGS [1]                 |   2.283    |   1.930   |    -1.378     |    -0.430     |
> | Controlled Decoding [2]  |   2.914    |   2.567   |    -1.777     |    -0.843     |
>
> ---
>
> > W2: The token-based MDP formulation is largely derived from existing work [3]. The main innovation of this paper lies in using the log-likelihood difference between two small models for guidance based on the existing formulation.
>
> The key contributions of this work are twofold:
>
> 1. **Formulation Contribution**: We frame alignment as a KL-constrained search and, more importantly, **decouple** the reference model that parametrizes the implicit reward from the reference model that constrains the search space, **enabling training-free alignment**. While some analyses are inspired by [3] (**which is actually concurrent to our work**), this decoupling framework that allows for practical training-free alignment is novel.
>
> 2. **Empirical Contribution**: Beyond demonstrating the effectiveness of training-free alignment, our framework shows consistent weak-to-strong generalization, namely enhancing strong models with weak guidance (see Figure 3, Figure 4, and our response to Reviewer [mz4P](https://openreview.net/forum?id=dOJ6CqWDf1&noteId=aC7BMDbknK)'s W4). **The empirical finding that our method achieves weak-to-strong generalization at test time is nontrivial (as other test-time baselines fail) and addresses a significant research question [4].**
>
> ---
>
> > Q1: What is the difference between the formulation in section 4.1 and the results in [3]?
>
> Please see our response to W2.
>
> ---
>
> If these answers do not fully address your concern, we are more than willing to offer additional clarifications.
>
> [1] Khanov, Maxim, Jirayu Burapacheep, and Yixuan Li. "ARGS: Alignment as reward-guided search." arXiv preprint arXiv:2402.01694 (2024).\
> [2] Mudgal, Sidharth, et al. "Controlled decoding from language models." arXiv preprint arXiv:2310.17022 (2023).\
> [3] Rafailov, Rafael, et al. "From $r$ to $Q^*$: Your Language Model is Secretly a Q-Function." arXiv preprint arXiv:2404.12358 (2024).\
> [4] Burns, Collin, et al. "Weak-to-strong generalization: Eliciting strong capabilities with weak supervision." arXiv preprint arXiv:2312.09390 (2023).

---

### Author Rebuttal · Authors · 2024-08-05

We are very thankful for the positive feedback from the reviewers. In this supplementary PDF, we include two additional empirical results: Figure 1: **inference costs analyses** [reviewer [mz4P](https://openreview.net/forum?id=dOJ6CqWDf1&noteId=aC7BMDbknK), W1] and Figure 2: **chunk-level PPO ablations** [reviewer [2D8T](https://openreview.net/forum?id=dOJ6CqWDf1&noteId=fzguH9HxKe), W2]. Please see the supplementary PDF and our individual responses to reviewer [mz4P](https://openreview.net/forum?id=dOJ6CqWDf1&noteId=aC7BMDbknK) and reviewer [2D8T](https://openreview.net/forum?id=dOJ6CqWDf1&noteId=fzguH9HxKe) for details.

---

### Decision · Program_Chairs · 2024-09-25

**Decision:**

Accept (poster)

**Comment:**

This is a solid paper that proposes to align larger language models by using the difference in logits from smaller (tuned vs. untuned) models. The idea is interesting, and backed up by solid empirical experiments. There were some initial concerns as to the use of stronger baselines (in particular, those that incorporating search during decoding), as well as its position relative to related work. These have been largely addressed through the rebuttal.

After reading the paper, I recommend that the authors further discussion the limitations of the proposed method (e.g., this method clearly would not work if the difference in capabilities of the two models is too large (e.g., n-gram model vs. a neural LM).